



# Future changes in water availability: Insights from a long-term monitoring of soil moisture under two tree species

Nikol Zelíková[1,2], Jitka Toušková[1], Jiří Kocum[1,3], Lukáš Vlček[1], Miroslav Tesař[1], Martin Bouda[4,5], Václav Šípek[1]

[1] Institute of Hydrodynamics of the Czech Academy of Sciences, Pod Patankou 30/5, Prague, 160 00, Czech Republic
[2] Department of Water Resources and Environmental Modelling, Faculty of Environmental Sciences, Czech University of Life Sciences Prague, Kamýcká 129, Praha-Suchdol, 165 00, Czech Republic
[3] Department of Physical Geography and Geoecology, Faculty of Science, Charles University in Prague, Albertov 6, Prague, 120 00, Czech Republic
[4] Department of Plant Ecophysiology, University of Hohenheim, Garbenstraße 30, Stuttgart 70599, Germany
[5] Institute of Botany of the Czech Academy of Sciences, Zámek 1, Průhonice, 252 43, Czech Republic

*Correspondence to*: Václav Šípek (sipek@ih.cas.cz)

**Abstract.**

Vegetation interacts with both soil moisture and atmospheric conditions, contributing to water flow partitioning at the land surface. Therefore, both climate and land cover changes impact water resource availability. This study aimed to determine the differential effects of climate change on the soil water regime of two common Central European forest types: Norway spruce (Picea abies L.) and European beech (Fagus sylvatica L.) stands. A unique dataset, including 22 years (2000–2021) of measured soil water potentials, was used with a bucket-type soil water balance model to investigate differences in evapotranspiration and groundwater recharge both between the forest types and across years. While long-term column-20   averaged pressure head indicated drier soil at the spruce site overall, this was driven by the wettest years in the dataset. Seasonal and interannual variability of meteorological conditions drove complex but robust differences in flow partitioning between the forest types. Higher snow interception by spruce (27 mm season[-1]) resulted in drier soil below the spruce canopy in the cold season. Higher transpiration by beech (70 mm season[-1]) led to increasingly drier soils over the warm seasons. 25   Low summer precipitation inputs exacerbated soil drying under beech as compared to spruce. Estimated summer recharge was lower under beech (25 mm season[-1]) due to its lower transpiration. The difference was more pronounced (over 40 mm season[-1]) during wetter summers. These suggest that expected trends in regional climate and forest species composition may interact to produce a disproportionate shift of recharge from the summer to the winter season.

## 1 Introduction

Advancing ecohydrological process understanding within a non-stationary state of the Earth system requires long-term observations of variables with direct mechanistic relevance. A key variable suffering from a noted information gap that hinders our understanding of land-atmosphere interactions is water potential or hydraulic head in soil and plants (Novick et





al. 2022). Soil moisture status integrates the fluxes of the entire hydrological cycle and in turn exerts significant control over key Earth system processes (Legates et al., 2011; Humphrey et al. 2021). As water potentials directly drive the soil-plant-atmosphere water flows that are tightly coupled with other land-atmosphere fluxes, addressing the water potential information gap offers a promising pathway to resolving major remaining uncertainties in ecosystem functioning that are attributed to soil moisture (Trugman et al. 2018, Green et al., 2019).

After centuries of relative climatic stability (Brázdil et al., 2022), a clear rise in average and maximum air temperatures has been affecting Central Europe since the last part of the 20th century (Zahradníček et al., 2020). Increased air temperature has induced higher atmospheric water demand (Možný et al., 2020), which can trigger a shift from energy to water limitation of evapotranspiration. Although long-term annual precipitation sums have not changed in the past (Brázdil et al., 2021) and are not expected to change significantly in near future (Svoboda et al., 2016), the occurrence of seasonal precipitation deficits causing severe soil drought is expected to strongly increase in upcoming decades (Hari et al., 2020). One of the less well understood consequences of climatic changes is a shift in forest species composition, which has the potential to affect water fluxes in the soil-plant-atmosphere system (Maxwell et al., 2018).

The two most frequent tree species in central European forests are represented by beech (Fagus sylvatica L.) and spruce (Picea Abies L.). As spruce thrives in colder and moisture-rich conditions, its stands in lower altitudes are often being replaced by beech (Daněk et al., 2019). This climate-induced transformation of mid-altitude forests brings consequences. Each of these species has distinctive physiological and architectural properties that are reflected in their specific strategies and characteristics diversely affecting soil water fluxes, water storage, and thus the soil water regime (Schume et al., 2004). They differ namely in the rate of interception (Savenije, 2004), rooting depth (Jost et al., 2012), and stomatal control during dry periods (Gebhardt et al., 2023). However, studies comparing soil moisture regime under both tree species provide ambiguous results. Schume et al. (2004) and Šípek et al. (2020) reported greater soil profile drying during the vegetation season at beech sites. By contrast, Schwärzel et al. (2009), Rötzer et al. (2017), and Kuželková et al. (2024) observed greater soil drying under spruce than under beech. These contrasting results may be partly due to differences in soil hydraulic properties at the compared sites, but they likely arise directly from the limited temporal extent of the studies. The periods of the observation range from one day (e.g., Jost et al., 2012) to several years (Schume et al., 2004; Schwärzel et al., 2009; Zucco et al., 2014; Korres et al., 2015; Huang et al., 2016; Rötzer et al., 2017). The longest periods of analyses so far lasted from 4 to 5 years (Wang et al., 2018; Šípek et al., 2020; Gebhardt et al., 2023). The results of short-term studies are difficult to interpret as they provide only a partial view of the role of individual water fluxes. They are limited by the variability of climatic conditions during the study period. Moreover, short-term studies also cannot capture any short-term changes in the characteristics of droughts, such as higher temperatures (Groissord et al., 2021) and flash droughts (Qing et al., 2022) and therefore their second-order effects via the given species. Hence, the availability of a long-term data series is crucial not only to observe trends, but it is a tool to better understand processes and natural variability in a period of changing climate and land cover (Huntingford et al., 2014; Milly et al. 2015).





This study aims to contribute to disentangling the effects of climate and forest composition on water availability in future conditions. We focused on the impact of two forest types, Norway spruce and European beech, on the soil water regime in a mid-altitude experimental catchment in Bohemian Forest, Czechia. The study benefits from the unique 22-year-long dataset of measured soil water potential under the two tree species. This allows us to obtain a comprehensive insight into the issue

across a wide range of climatic variables. The main goals of our study include: (1) to analyse seasonal differences in measured soil water potential between the two forest types, (2) to estimate the soil water balance components (evapotranspiration and drainage) at the two sites using a process-based soil water balance model, and (3) to determine the main climate dependency of the soil water regime under both tree species.

## 2 Data and Methods

### 2.1 Study site

The Liz experimental catchment, Czechia (49°04′N, 13°41′E) (Fig. 1), served as the experimental area for this study. It is located in the Bohemian Forest on the border between Czechia and Germany. The catchment area is approximately 1 km². Its elevation ranges from 828 m a. s. l. (the outlet) and extends to an altitude of 1,070 m a. s. l. The climate is humid continental with an average annual air temperature of 7.2 °C and an average annual precipitation of approximately 847 mm. The

monthly average maximum temperature is 16.5 °C (July), and the minimum is −1.9 °C (January). The maximum precipitation occurs in July (104.6 mm), and the minimum occurs in November (47.6 mm). The annual potential evapotranspiration (PET) determined by the air temperature-based method (Oudin et al., 2005) is 595 mm. The annual runoff height from the catchment is approximately 360 mm, representing ~40% of the total precipitation. All meteorological and hydrological data were collected for the period 2000–2021.

Crystalline bedrock in the catchment only allows water circulation in the weathered zone and does not communicate with adjacent catchments. Therefore, the hydrological catchment corresponds well to the hydrogeological catchment (Hrkal et al., 2009). The majority of the area is covered by nearly pure spruce forest, with a dominance of 120−140-year-old Norway spruce (*Picea abies* L.) (> 85% of the canopy cover). In several places, the spruce forest is penetrated by 100−120-year-old beech stands (*Fagus sylvatica* L.).

Two experimental sites within the Liz experimental catchment were chosen for this study: one with Norway spruce (*Picea abies* L.) and the other with European beech (*Fagus sylvatica* L.),. The elevation difference between the two sites is approximately 30 m: the spruce site is located at 855–860 m a.s.l., and the beech site is located at 885–890 m a. s. l., both with a slope of 7.5° and an eastern aspect. Both spruce and beech canopies tend to suppress understory vegetation, which was accordingly absent at both sites (Fig. 1). The leaf area index was measured throughout the 2022 season on a monthly

basis and showed a seasonally stable value with an average of 3.7±0.5 in the spruce site and seasonally variable values in beech ranging from 1.1±0.2 at the beginning and end of vegetation season (May and September) to 4.7±0.5 in the middle of



the growing season. A visual inspection of the root depth distribution (when excavating the soil) revealed that the roots were present only in the upper 40 cm of the spruce site and down to 100 cm of the beech site.

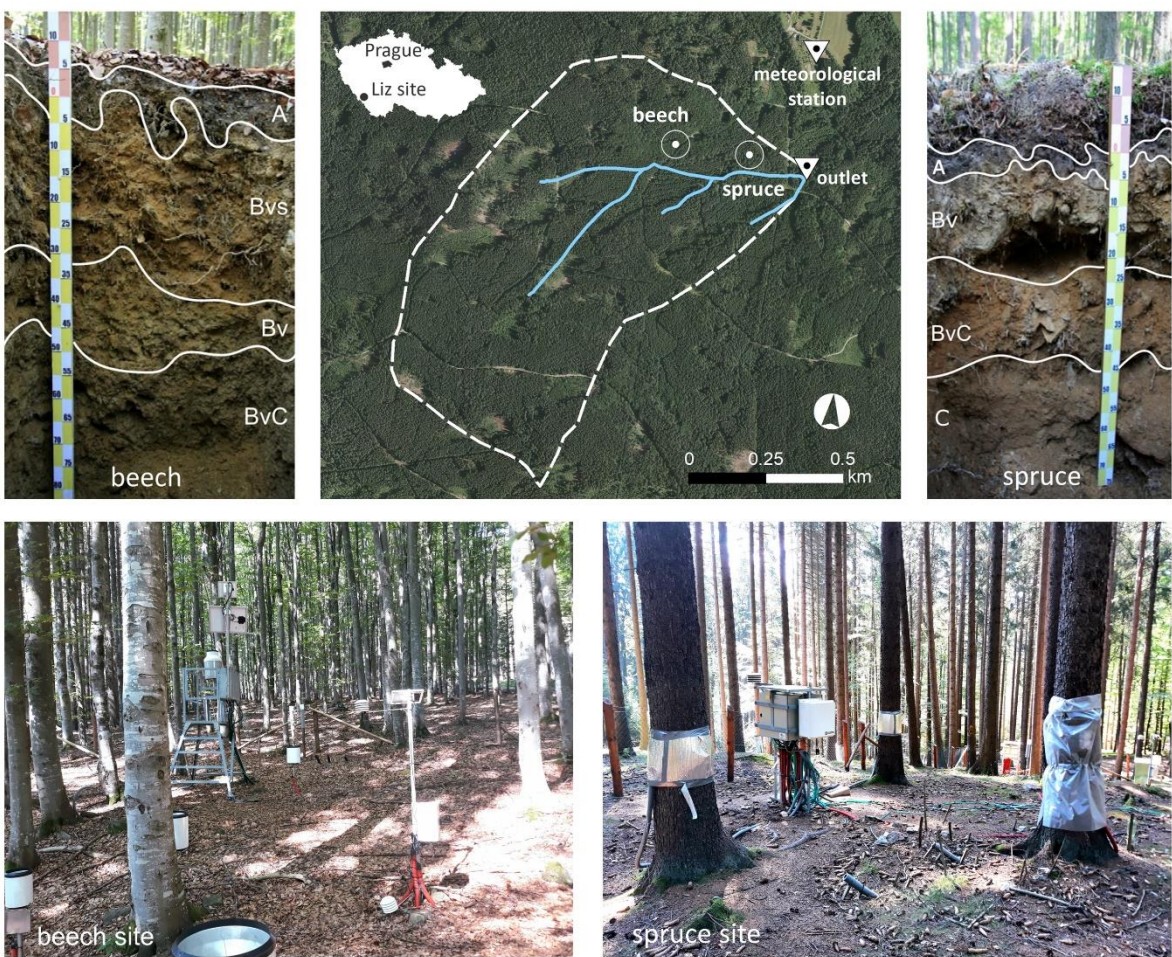

**Figure 1: Overview of the experimental site (© CUZK 2024) and soil profiles (© Přemysl Fiala).**

The soil at both sites can be classified as moderately deep loamy sand dystric Cambisol (IUSS, 2015), with an average soil depth of approximately 100 cm. The percentages of sand-silt-clay fractions are 73.2%–24.2%–2.6% at the spruce-covered site and 80.2%–18.1%–1.7% at the beech-covered site. The soil water permeability is relatively high ranging from 518
cm.day$^{-1}$ at the bottom of the soil profile to 1700 cm.day$^{-1}$ in the topsoil horizon. The humus A horizon (0–10 cm), together with surface organic horizon O (5–10 cm thick at beech stand and 10–15 cm at spruce stand), is followed by a Bvs/v horizon (down to 50 cm at beech site and to 30 cm at spruce site) and finally by a BvC horizon with a significant amount of larger than sandy particles (>50%). Both soil profiles are presented in Fig. 1.



## 2.2 Field measurements

The meteorological variables used in this study were air temperature (Fiedler RV12/RK5, Czech Republic) and precipitation (Meteoservis MRW 500, Czech Republic), which were measured at 10-minute intervals during the entire twenty-two-year period (2000−2021). The meteorological station is located circa 400 meters away from two experimental plots outside the forest. Moreover, the experimental catchment is instrumented with discharge and groundwater level measurements. Discharge was also measured at the 10-min time step, and the groundwater level was recorded manually every week

throughout the entire investigated period. The snow water equivalent (SWE) was measured manually three times per week. Soil water potential data were acquired from permanently installed soil tensiometers (Adolf Thies GmbH, Germany) measuring pressure heads at five depths (15, 30, 45, 60 and 90 cm). Soil water potential data were recorded manually three times a week during the vegetation season (mid-May to mid-October) from 2000 to 2021. The measuring range of these tensiometers included pressure heads ranging from 0 cm to −865 cm (−85 kPa). One to four tensiometers were available for

each measuring depth at each site, and the single value for a particular depth was taken as their average. The average soil column pressure head was estimated as a weighted mean of five soil layers (each represented by one measurement depth). The soil layers were separated by depths of 22.5, 37.5, 52.5 and 75 cm between two adjacent tensiometer measurement points. The soil profile was considered to have a uniform depth of 100 cm. The measured pressure heads were used to determine differences in soil water regimes between the stands, as they better demonstrated the stands' behavioural

differences during dry conditions, which were of interest to the study.

## 2.3 Soil water balance model

The conceptual model used in this study was a modified form of the soil water balance model (SWBM), developed by Brocca et al. (2008, 2014). Several widely used hydrological models use similar bucket/reservoir modelling approaches for the determination of soil water regimes (e.g., the Soil Water Assessment Tool (Arnold et al., 2012), the HBV model (Seibert

and Vis, 2012) or the VIC model (Liang et al., 1994)). The modification for this study is based on the replacement of the infiltration parameter (the Green-Ampt equation) by throughfall $(P_{TF})$, as surface runoff is not generated in the experimental catchment and all water directly infiltrates into the soil. Therefore, the following soil water balance Eq. (1) was used:

$$\frac{d\Theta(t)}{dt} = P_{TF}(t) - S(t) - D(t) \tag{1}$$

where $P_{TF}(t)$ is the throughfall (mm day$^{-1}$), $S(t)$ is the actual evapotranspiration rate (mm day$^{-1}$) and $D(t)$ is the drainage rate

(mm day$^{-1}$). The Eq. (2) for $P_{TF}(t)$ is given as:

$$P_{TF}(t) = P_{OAR}(t) - P_{INT}(t) \tag{2}$$

where $P_{OAR}$ represents the measured open area precipitation (mm day$^{-1}$) and $P_{INT}$ is the estimated interception (mm day$^{-1}$) for a given location. Spruce interception in the summer season (May to October) was estimated based on the deduction of the





interception capacity from every single precipitation event. The interception capacity of 2.2 mm was derived by Kofroňová
et al. (2021) for the same experimental site. In the case of beech stands, the summer interception capacity was calculated
using a general formula by von Hoyningen-Hüne (1983) and Braden (1985) applying seasonal variation in the leaf area index
(LAI):

$$P_{INT} = a \cdot LAI \left( 1 - \frac{1}{1 + \frac{b \cdot P_{OAR}}{a \cdot LAI}} \right) \quad (3)$$

where $a$ is an empirical coefficient (-) and $b$ is the soil cover fraction (=LAI/3.0) (-). Daily values of LAI were acquired from
linear interpolation between monthly measured values (May−September) conducted by a LI-COR 2000 Plant Analyser in
2022 (Toušková et al., unpublished results). The calibration of $a$ parameter was performed so that the fraction of intercepted
precipitation was allowed to range between 15 and 20%, which is an ordinary interception loss of beech canopies (Gerrits et
al., 2010). For the winter season (November to April), linear regression functions linking open area snow water equivalent to
that below the forest canopy were used (Šípek and Tesař, 2014). The regression equations are based on the measured snow
water equivalents in the forest openings and below the spruce (Eq. 4) and beech (Eq. 5) canopies for a period of ten years
and are in the form:

$$SWE_{TF}(t) = SWE_{OAR}(t) \cdot 0.595 \quad (4)$$

$$SWE_{TF}(t) = SWE_{OAR}(t) \cdot 0.679 \quad (5)$$

where $SWE_{OAR}$ is the snow water equivalent (mm day$^{-1}$) in the open area and $SWE_{TF}$ is the snow water equivalent under the
forest canopy (mm day$^{-1}$).

Potential evapotranspiration (PET) was estimated using the Oudin et al. (2005) approach, which offers reliable estimates of
PET for long-term water balance studies in the Central European region (Toušková et al., 2024). Net longwave radiation was
estimated using the FAO56 approach with site-specific coefficient values (Kofroňová et al., 2019). The actual
evapotranspiration rate (comprising soil evaporation and plant transpiration) was then estimated based on the linear decrease
in its potential rate with decreasing effective soil water content as proposed by (Feddes and Rijtema, 1972) according to the
following Eq. (6) and Eq. (7):

$$S(t) = PET(t) \cdot \Theta_e \quad (6)$$

$$\Theta_e = \left[ \frac{\Theta_{(t-1)} - \Theta_r}{\Theta_s - \Theta_r} \right] \quad (7)$$

where $PET$ is the potential evapotranspiration (mm day$^{-1}$) and $\Theta_{e,r,s}$ are the effective, residual and saturated soil water
contents (mm), respectively. The drainage component $D(t)$ is a nonlinear function of $\Theta_e$:

$$D(t) = K_s \Theta_e^{3 + \frac{2}{\lambda}} \quad (8)$$





where $K_s$ is the saturated hydraulic conductivity (mm h$^{-1}$) and $\lambda$ is the pore size distribution index (-) linked to the textural structure of the soil layer, which was set to 0.5. In this case, the flow is assumed to be gravity driven, with drainage consisting of deep percolation.

## 2.4 Modelling procedure

The original SWBM does not include a snow module; hence, snow accumulation and snowmelt had to be considered first, as the experimental catchment lies in an area with regular snow cover. The degree-day method (Gupta, 2001) was chosen for this purpose because it has been proven to be efficient in the Central Europe (Girons Lopez et al., 2020). The governing parameters were calibrated separately for each winter season so that the input for the soil water model was as accurate as possible. The calibrated parameters included the snowfall correction factor ($SFCF$), two threshold air temperatures—one for snow to occur ($T_{snow}$) and the second for the snowmelt to begin ($T_{melt}$)—and the degree-day factor controlling the rate of snowmelt based on the air difference between the average daily air temperature and the threshold temperature ($DDF$). The measured values of the SWE were used for model calibration. The calibration was based on the minimisation of the RMSE between modelled and measured values using the genetic algorithm.

For the purposes of soil water balance modelling, the measured pressure heads were used to calculate the volumetric soil water content by means of the van Genuchten (1980) function. The function parameters were retrieved from the measured retention curves specific for each site and depth (see Table S1 in Supplementary material). For more information about the determination of the soil water retention curves, we refer to Šípek et al. (2020).

The parameters of the SWBM were calibrated with respect to the measured soil water contents at both sites. The calibration was based on minimising the RMSE by means of a genetic algorithm. The following parameters were used for calibration: saturated ($\theta_s$) and residual ($\theta_r$) soil volumetric water content and saturated hydraulic conductivity ($K_s$).

In addition to the minimisation of the RMSE, the model was calibrated with two additional requirements: (1) simulated drainage from both sites must equal approximately 360 mm y$^{-1}$, which is a value obtained from the long-term measured runoff from the area, and (2) beech summer interception loss will be within 15–20% of the open area rainfall, which corresponds to the range reported by Gerrits et al. (2010).





# 3 Results

## 3.1 Analysis of the measured soil water regime

The climate conditions of all investigated summer and winter seasons are depicted in Fig. 2. One wet (2002) and one dry year (2015) were chosen to demonstrate differences among pressure heads between the spruce and beech sites influenced by extreme meteorological conditions (Fig. 3 and 4). Dry and wet years were identified by analysing the soil moisture regime in terms of the vertical distribution of pressure heads (down to a depth of 100 cm) over the entire period (2000–2021) and sorting of specific years into soil wetness categories based on the seasonal distributions (from May to August) of the measured pressure heads.

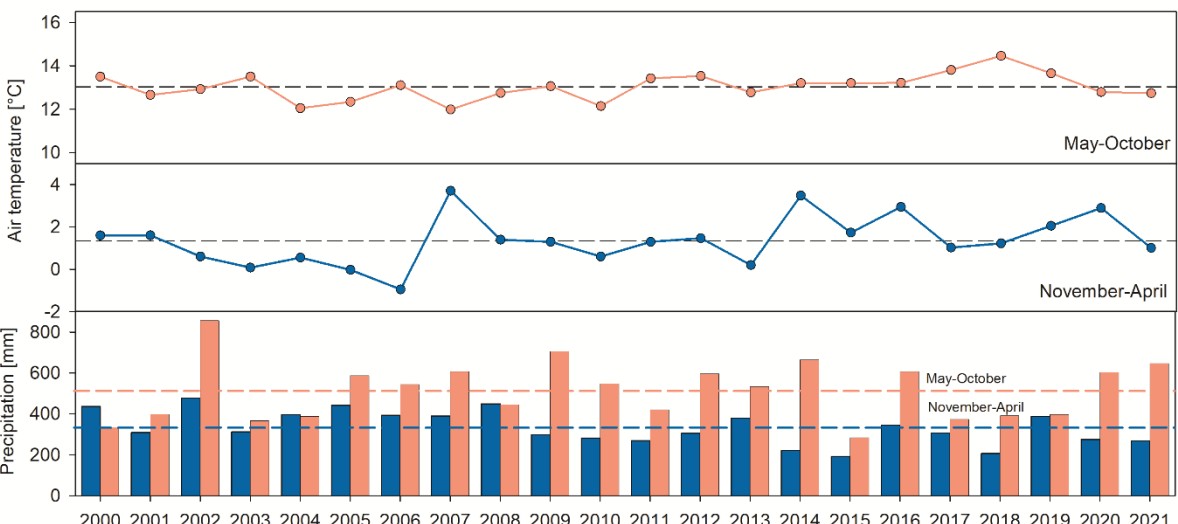

**Figure 2: Average air temperatures (upper two panels) and precipitation sums (bottom panel). The red columns represent the summer seasons (May–October), and the blue columns represent the winter seasons (November–April). Dashed lines represent season average values.**

## 3.1.1 Vertical distribution of pressure heads

Pressure-head values were slightly higher at the beech site, with a median of −155 cm compared to −255 cm for the spruce site. However, despite the higher median pressure-head values recorded at the beech site, the occurrence of low pressure-heads ($< -500$ cm) was more frequent here (as reflected by more extended boxes towards lower pressure heads in Fig. 3). This phenomenon became more pronounced with increasing depth. All recorded pressure head values at the spruce and beech sites, displayed in Fig. 3, exhibited only slight differences in terms of column average values. Differences in the vertical distribution of pressure heads were visible, namely, in the topsoil layer (depth of 0–15 cm), where soil under spruce attained lower pressure head values than that under beech. However, the overall depth distribution of the pressure heads was





relatively uniform under spruce. In contrast, the pressure head depth distribution under beech trees exhibited greater propensity to dry in the bottom soil layers, although the median values were also uniformly distributed with depth. The beech site, despite having higher pressure heads on average, was therefore more susceptible to more intensive drying than the spruce site, as indicated by greater pressure head variance.


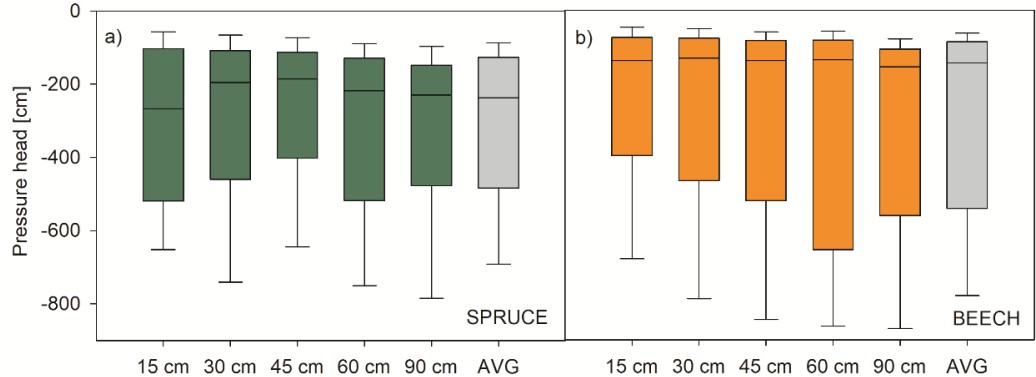

**Figure 3: Ranges of all recorded pressure heads in the spruce (a) and beech sites (b) for all measuring depths and soil column averages (2000–2021).**

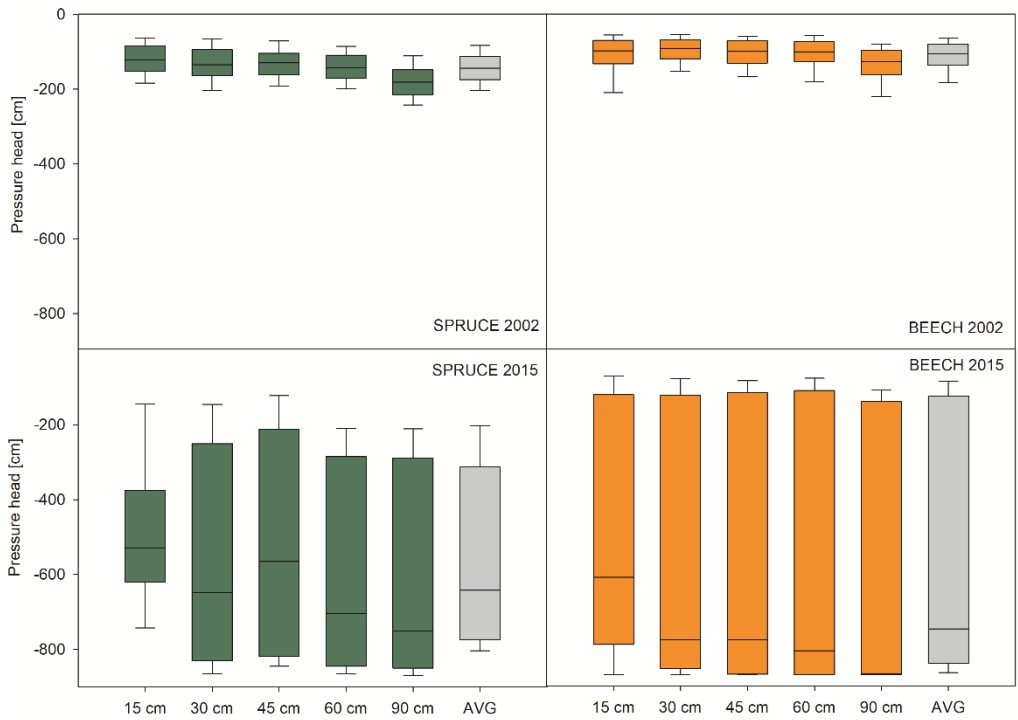


**Figure 4: Ranges of all recorded pressure heads in selected wet (2002) and dry (2015) years at the spruce and beech sites.**





The differences between the beech and spruce site were nearly indistinguishable during the wet years but the soil under beech was noticeably drier in dry years (see examples of the wet year 2002 and dry year 2015 in Fig. 4). Although the

differences among the sites were small in wet years, lower pressure heads were observed at the spruce site at all depths (circa 30 cm pressure head difference between beech and spruce for the whole soil column). In contrast, during the dry year of 2015, the soil under spruce site was wetter (attained a higher column average median pressure head) than at the beech site. Pressure heads lower than −850 cm were observed at a five to tenfold greater frequency at the beech site compared to spruce site depending on the measuring depth. Hence, the differences in pressure heads might be even greater, as the tensiometer

data reached their limit more frequently at the beech site than at the spruce site; thus, even lower pressure heads were likely to occur at the beech site. If the number of dry years increase in the future, the soil under beech will become drier during the vegetation seasons.

### 3.1.2 Soil wetness categories

The measured pressure heads were divided into four soil wetness categories based on their typical seasonal development.

The evolution of average pressure heads for each month of the summer season over the measured period (2000–2021) is depicted in Fig. 5. At both sites, a similar pattern of decreasing pressure heads from the onset of the summer season can be observed. However, there are noticeable differences between the two sites. At the beginning of every summer season (May), the spruce site attained lower pressure head values than did the beech site (the average difference in pressure heads was 130 cm). Typically, as the season progresses, the pressure heads at the beech site decrease more than those at the spruce site.

However, this was not valid for the wet seasons of 2002, 2005, 2020, and 2021, when spruce retained lower pressure heads throughout most of the season (see Fig. 5), as no precipitation deficit was observed (category A). For those seasons, the difference between the two sites was negligible, with their average values fluctuating between −100 and −200 cm. In the other few years, when above average precipitation seasonal sums were attained (category B, including the years 2006, 2010, 2012, 2014, and 2016), there was only one single event when the beech site attained lower pressure heads (below −400 cm),

which was usually ended by rainfall higher than 50 mm day$^{-1}$. In contrast, in the periods with below average precipitation, the pressure head decreased more pronouncedly at the beech site for a significant part of the summer season (category C included, e.g., years 2007, 2012 or 2019, as shown in Fig. 5). With even more prominent precipitation deficits (in 2003, 2008, 2015 and 2017), the beech site was the first and often only site to reach the tensiometer measurement limit of –865 cm (category D) - up to ten times more often than the spruce site, especially in the bottom soil layers. Real pressure head values

were likely significantly lower. As lower pressure heads cannot be recorded at the beech site with tensiometer measurements (measuring limit was reached) and pressure heads at the spruce site only seldom approached this limit, the differences between both sites was higher than documented by sensors. The effect on our analysis was likely insignificant as the implied differences in the amounts of water retained would be rather small. By the end of the season, pressure head values slowly increased, with beech still maintaining lower pressure head values than spruce.






**Figure 5: Daily precipitation (P) (black columns) and soil column average pressure heads at beech (orange line) and spruce (green line) sites in all investigated years divided into four wetness categories (A–D) defined by pressure head values. The red dashed line represents the pressure head of −400 cm used for the division of categories A and B.**





### 3.2 Modelling of evapotranspiration and drainage

**3.2.1 Model calibration**

The modified SWBM model was used to obtain evapotranspiration and drainage fluxes over a period of twenty-two years (2000–2021) at both spruce and beech sites. The entire period of available data was used for model calibration.

First, the degree-day snow accumulation/melt routine parameters were calibrated against the observed snow water equivalents. The RMSE values were 7.1 mm (beech) and 9.5 mm (spruce), which are in accordance with Šípek and Tesař (2017), who modelled snow cover dynamics from 2009 to 2014 and reached an RMSE value of 9.1 mm in a spruce stand. An example of the modelled cumulative snow precipitation fitted to the measured SWE is shown in Fig. S2a in supplementary material.

Second, the remaining model parameters were calibrated against the measured soil water content at both the beech and spruce sites. The resulting SWBM efficiencies in terms of the RMSE were 2.5% and 2.8% for the spruce and beech sites, respectively. The modelled long-term drainage was 353 mm year$^{-1}$ for beech and 365 mm year$^{-1}$ for spruce. The average annual discharge for the experimental Liz catchment was 360 mm, which was very close to the modelled values. The final parameters of the SWBM ($\theta_s$, $\theta_r$, $K_s$, $\lambda$) for each site are documented in Table 1. Examples of modelled and observed VWC are depicted in Supplementary material (Fig. S2b).

**Table 1. Calibrated soil water balance model parameters**

|        | $a$  | $\theta_s$ | $\theta_r$ | $K_s$ | $\lambda$ | *RMSE* |
|--------|------|------------|------------|-------|-----------|--------|
| Spruce | -    | 514.4      | 79.8       | 165.9 | 0.50      | 2.5 %  |
| Beech  | 0.50 | 453.0      | 0.0        | 21.0  | 0.50      | 2.8 %  |

**3.2.2 Simulated Water balance**

The total actual evapotranspiration (AET) and drainage attain similar values at both plots on average. The major differences lie in rates of transpiration/soil evaporation (T+E) and interception. Transpiration is higher in the beech forest and drainage in spruce forest. The total AET is approximately 540 mm season$^{-1}$, and the drainage is between 350 and 360 mm season$^{-1}$ (Table 2). Beech reaches almost 100 mm higher T+E, and similarly, spruce reaches this level in the case of interception. The resulting AET values therefore do not differ greatly from each other because T+E and interception tend to compensate for each other between stands, which is hence also reflected in similar drainage.

Even though the winter seasons are characterised by lower precipitation sums than the summer seasons (approximately 1/3 of the annual precipitation), the spruce forest had, on average, a higher rate of interception (133 mm season$^{-1}$) due to defoliated beech forest (106 mm season$^{-1}$) (Table 2). However, from the AET perspective, the difference in interception is partially alleviated by slightly higher transpiration and soil evaporation under the beech canopy at the beginning and end of the winter season (14 mm season$^{-1}$). Nevertheless, the interception rate and winter transpiration at the spruce site resulted in





a lower amount of water available for infiltration and therefore a lower modelled soil water content during the winter months. The drier soil in spruce forests regularly represents an initial condition for the summer season. A higher soil water content below the beech canopy was a reason for higher modelled drainage during the winter season at the beech site (by 12 mm season$^{-1}$ on average).

**Table 2: Modelled soil water balance components (mm) at the spruce and beech sites. T+E represents transpiration and soil evaporation from the soil column. AET stands for actual evapotranspiration.**

|  | 2000–2021 | | WET 2002 | | DRY 2015 | | Winter season | | Summer season | |
|---|---|---|---|---|---|---|---|---|---|---|
|  | SPR | BEE | SPR | BEE | SPR | BEE | SPR | BEE | SPR | BEE |
| Precipitation | 901 | | 1398 | | 499 | | 340 | | 561 | |
| T+E | 261 | 345 | 279 | 376 | 221 | 297 | 46 | 62 | 213 | 282 |
| Interception | 275 | 204 | 353 | 256 | 201 | 139 | 133 | 106 | 143 | 103 |
| AET | 536 | 549 | 632 | 632 | 422 | 436 | 179 | 168 | 356 | 385 |
| Drainage | 365 | 352 | 702 | 673 | 145 | 161 | 162 | 174 | 205 | 180 |

In the summer, transpiration flux significantly affected the water balance at both sites as it was noticeably higher in the beech forest (see Table 2). The interception pattern of both stands was preserved, with spruce having higher interception 305 (142 mm season$^{-1}$) than beech (103 mm season$^{-1}$). The differences in the soil water content were therefore caused by the transpiration in the beech stands (by 70 mm season$^{-1}$). Hence, soil under spruce trees retained (with the ongoing summer season) more water than soil under beech trees, where soil moisture was more effectively used for higher transpiration of beech trees, especially during dry spells. The wetter soil under spruce (in the majority of summer seasons) resulted in higher drainage by 25 mm season$^{-1}$ on average.

**3.3 Climate-induced soil water regime and soil water fluxes**

 Figure 6 shows the relative rankings of individual study years according to snow cover duration, air temperature, summer precipitation (May-October), and their classification into the four wetness categories according to the resulting pressure head dynamics, shown in Fig. 5.

The dominant factor controlling the soil water regime in the growing season was the amount of summer precipitation. A 315 significant soil moisture deficit could develop even following a winter with abundant snow. Fig. 6 clearly shows the direct





link between pressure head and summer precipitation, where lower pressure heads are linked mainly to years with lower seasonal precipitation, and higher pressure heads are linked to years with abundant precipitation. The correlation coefficient between summer precipitation and soil moisture regime category was 0.80 (significant at 0.05 probability level). Two marginal categories (A and D) were always linked to specific climatic conditions (see Fig. 6). Category A, denoting wet soil

(hence small differences between beech and spruce sites), was always determined by above average precipitation amounts and below average air temperatures observed in the summer season. Category D, representing the very dry soil moisture regime, was always accompanied by low observed precipitation amounts in the summer season. Two middle categories (wetter B and drier C) tend to be connected primarily with above (in the case of B) and below (category C) average precipitation sums. The influence of preceding winter snow cover and summer season air temperatures was ambiguous, as

seen in the frequently strongly mismatched placement of particular seasons along these axes in Fig. 6, compared to the resulting soil wetness category. The correlation coefficient with soil moisture regime were 0.30 and 0.08 for summer air temperature and snow cover duration, respectively. Higher correlation coefficient was also observed for the summer vapour pressure deficit (VPD) attaining the value of 0.61 (not shown in the Fig. 6).

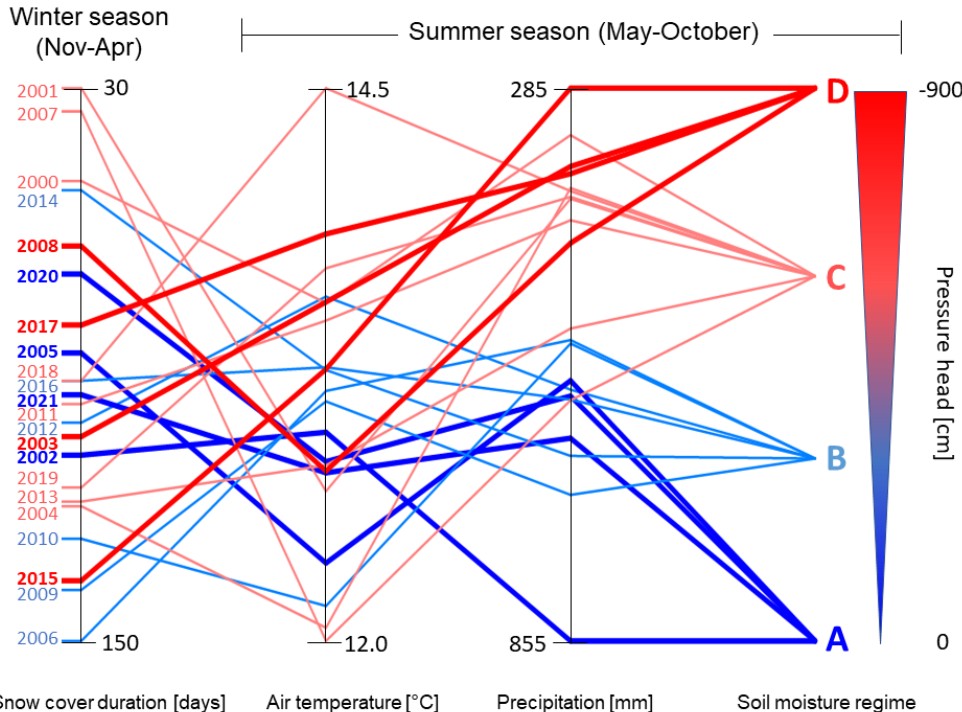

**Figure 6: Average air temperature and precipitation sums for each summer season (represented by one horizontal line) encompassing the preceding winter snow cover duration. Each season is ultimately linked to a specific wetness category (A–D), as shown in Fig. 5.**



The most pronounced deviations from the observed link between summer precipitation sums and the soil moisture regime
were in the 2013 and 2007 seasons, with above average precipitation but a drier soil moisture regime. This was caused by a
near absence of snow cover observed in the winter of 2006/2007, accompanied by the highest recorded winter air
temperature (Fig. 2), and by the extreme floods in 2013, when the catchment received 1/3 of all summer precipitation in June
but saw below average precipitation amounts during the rest of the season. These two factors caused a drier soil moisture
regime even when above average precipitation sums were recorded. These results therefore document how different rainfall
conditions influence the development of soil moisture content and the different behaviours of beech and spruce in vegetation
season (Fig. 5).

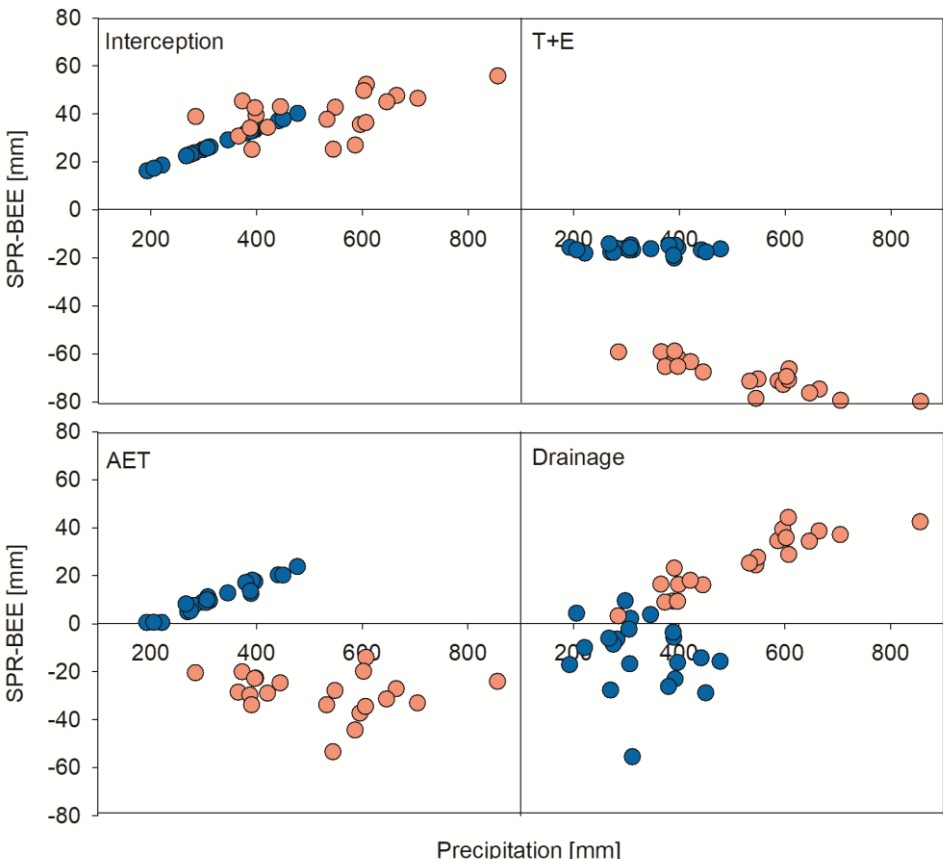

**Figure 7: Differences between spruce and beech modelled soil water fluxes (AET, D) during summer (May to**
**October, orange colour) and winter (November to April, blue colour) in relation to precipitation. AET can also be**
**divided into INT and T+E (upper panel).**

Seasonal precipitation also had a major influence on the differences in particular water fluxes between beech and spruce sites
(Fig. 7). Differences in all fluxes could be positively or negatively related to seasonal precipitation sums with the exception
winter transpiration and soil evaporation (T+E). The differences in winter/summer interception, winter actual



evapotranspiration (governed mainly by interception) and summer drainage increased with increasing precipitation. By contrast, summer transpiration and soil evaporation, summer actual evapotranspiration (governed by transpiration) and winter drainage were negatively related to precipitation sums. The largest absolute differences in water fluxes between the stands were recorded during wet summer seasons. The most pronounced discrepancies were in the rates of transpiration and

soil evaporation (higher for beech plots; up to 80 mm season[-1]), summer interception (higher for spruce plots; up to 55 mm season[-1]) and drainage (higher for spruce plots; up to 45 mm season[-1]). The lowest differences occurred during the dry winter seasons. The differences in the winter seasons were generally less prominent, usually below 40 mm season[-1].

## 4 Discussion

**4.1 Vegetation and climate interactions with soil moisture regime**

Our unique 22-year long dataset of measured pressure heads enabled robust comparisons of the soil water regime between dry and wet years, allowing modelled soil water fluxes under beech and spruce canopy to reveal the interactions between forest cover, climate, and soil moisture. We found that the differences in soil moisture regime under beech and spruce canopy were strongly dependent on the seasonal precipitation sums. The partitioning of the water fluxes in both stands was

driven by different rates of interception (higher at the spruce site) and transpiration and soil evaporation (higher under beech). The average annual AET was fairly similar, but discrepancies exist in individual seasons and years. Differences in winter soil moisture regime were determined mainly by the higher interception of the spruce canopy, which resulted in higher pressure heads under beech causing more drainage compared to the spruce site. When the precipitation in the following summer season was high, only minor differences in pressure heads were recorded between stands, even though the

spruce site maintained slightly lower pressure heads throughout most of the season (as a winter season legacy effect). The resulting differences remained small as the higher interception of spruce did not exceed the higher rate of transpiration of beech in the vegetation season.

As the growing season advanced, transpiration became an increasingly important factor in the soil moisture regime. The balance between interception and transpiration/soil evaporation resulted in greater drainage under the spruce canopy. In

seasons with prominent precipitation deficits, the soil at the beech site consistently dried out more than at the spruce site. This can be explained by species-specific plant hydraulic traits. Beech has a wider and deeper rooting pattern and thus soil volume and water in its root zone, especially at greater depths (Čermák et al., 1995; Schwärzel et al., 2009; Gebauer et al., 2012). Beech also has a greater tissue-specific hydraulic conductance due to favourable anatomical and morphological traits, allowing it to supply leaves with water more efficiently at a given root-zone water potential (Tyree & Zimmermann, 2002).

As a result, beech behaves more anisohydrically, maintaining transpiration rates in the face of drier soils, in contrast to the more isohydric spruce, whose lower ability to supply water to its foliage requires it to restrict transpiration earlier as the soil dries out (Čermák et al., 1995; Zweifel et al., 2002; Schume et al., 2004; Hochberg et al., 2017; Gebhardt et al., 2023). Also,





higher evaporation from soil and litter under beech stands contributes to overall evapotranspiration, as reported by Schwärzel et al. (2009) and Floriancic et al. (2022). Additional factors possibly affecting differences in soil water regimes include
lateral flow, which is reportedly more common at beech sites (Jost et al., 2012), and root water redistribution (Burgess et al., 1998). In dry summers, the drainage remained higher under spruce canopy, although the difference between the stands decreased as the difference between interception and transpiration declined.

Previous studies provide contradictory results on soil drying under beech versus spruce canopies. While Schume et al. (2004) and Šípek et al. (2020) observed drier soil during the growing season under a beech canopy, Schwärzel et al. (2009) found
the opposite. In the latter case the more prominent drying under spruce was attributed to the nonuniform and rocky soil compared to beech site. Rötzer et al. (2017) and Kuželková et al. (2024) also reported drier soil under spruce but these studies covered only the upper part of the soil profile (0—30 cm). We found that differences in the total amount of water in the soil column between beech and spruce during dry periods are dominated by depths of 30 cm and more, where the greatest differences arise. Previous contradictory observations thus result from the limited scope of measurements over space or time.
This is the first study we know of to cover more than a few seasons and thus to provide a robust interannual comparison of soil moisture regime under beech and spruce canopies integrated over the entire soil column. Viewed at this scale, the comparison of soil moisture regimes proves to be precipitation dependent.

Our results have some clear implications for how combined changes to climate and forest species composition in Central Europe should be expected to impact forest water regimes. Climate change is expected to shift the intra-annual distribution
of precipitation totals from summer to winter. We found that higher winter drainage under the beech canopy is even more pronounced with increasing winter precipitation. In combination with the increasing representation of beech trees in Central Europe, this can generally lead to higher winter groundwater recharge and runoff. By contrast, summer season differences indicate lower future recharge from beech forests given their higher transpiration demand, which generally consumes soil water even during drought periods. Overall, our results suggest that ongoing trends in climate and forest composition are
pushing these forest systems from energy-limited towards water-limited states, at least on a seasonal basis.

Our finding that summer season temperature did not greatly affect the water balance may in part be due to the relatively low temperatures in the relatively high-elevation study catchment. Atmospheric moisture demand can be quantified as vapour pressure deficit (VPD), which is a strongly nonlinear function of air temperature (Groissord et al., 2021). Thus, lower elevation forests with higher summer temperatures will face disproportionately higher VPD. This factor may also increase in
importance disproportionately across the landscape with further climate warming. Given our findings, we would again expect the effects on local water balance to be stronger in beech rather than spruce stands, due to their relatively anisohydric transpiration, and to shift the state of these systems further towards water limitation.

**4.2 Measurement limitations**

As the measuring limit of the tensiometers is −865 cm (−85 kPa), pressure heads below this limit could not be recorded.
Some information was therefore lost, especially at the beech site where periods with a constant limit value were clearly





visible. However, for pressure heads lower than the measurement limit, the loss/gain of the volumetric water content corresponding to the unit change in the pressure head is very small (a 100 cm change in the pressure head accounts for less than 0.002 cm$^3$ cm$^{-3}$ of the change in the volumetric water content). The same rate was observed for a saturation to a pressure head of $-100$ cm, which is equal to 0.22 cm$^3$ cm$^{-3}$. Hence, the changes in pressure head concerning such low heads

have a negligible effect on the volumetric soil water content.

To encompass the influence of soil moisture spatial variability, 2 to 5 tensiometers were used at each depth. As the standard errors of precipitation measurements are 10% in summer and 40% in winter, it can be assumed that these measurements of precipitation can also be biased due to wind eddies around the rain gauge and deposited precipitation (Dingman, 2015). Even though the study sites are located close to the rain gauges (<500 m) and we also checked the open area rainfall data with the

raingauges located in the forest, there were occasional episodes in the data where the volumetric water content did not match to the volume measured rainfall, which resulted in a few errors in the soil moisture modelling, especially of the rises in the volumetric soil moisture content.

### 4.3 Modelling limitations

Observations from times when soil pressure heads were at or below the tensiometer measurement limit were not used to

constrain or evaluate the soil water balance model. The model was allowed to run freely below this limit, and the error statistics from these periods were not considered. Another issue arose from the noted episodes of rainfall over-/underestimation. Eliminating this bias did not allow model fitting during dry periods. As both issues affected periods with negligible water fluxes, neither was found to affect the long-term water balance. Finally, as shown in Cejpek et al. (2018) and Jačka et al. (2021), different vegetation species growing on the same soil type tend to change soil properties, whether

due to different root systems, soil biology or litter. Even though the soil parameters ($Ks$, $\Theta_{e,r,s}$) that were entered into the balance model have measured equivalents at each site, their values in this study are the result of model calibration.

Since the model validation was performed on the average annual discharge value measured for the entire watershed, which is mostly covered by spruce forest, it is possible that these values may not correspond with the discharge that might occur from the beech site alone. This might affect confidence in the balance components (drainage and actual evapotranspiration) at the

beech site as compared to the spruce site. However, the modelled high transpiration rates at the beech sites mostly follow from fitting to the high-resolution time series of measured local soil moisture data, which show lower values during the summer season compared to spruce, and simultaneous observations of no change in groundwater levels. The comparatively high transpiration rates of beech during the summer season were separately validated by measured sap flow (Brinkmann et al., 2016; Gebhardt et al., 2023), which is higher during the summer season in beech than in spruce, especially in dry periods.

Moreover, due to the absence of measured soil moisture data below the tensiometer measurement limit, it could be assumed that as soil moisture values could be even lower at beech sites, transpiration will be higher than estimated. To avoid such uncertainties in future research, detailed sap flow measurements might serve for model calibration, which could then show the values of actual evapotranspiration and drainage more precisely.



## 5 Conclusion

Gradual changes in species composition in Central European forests and the increasing frequency of drought episodes due to climate change raised interest in the possible impacts of these changes on the water balance of forested catchments. Based on long-term soil water potential measurements, four categories of soil water regime at beech and spruce stands were determined. Generally, the drier the summer season was, the drier the soil was under the beech canopy compared to spruce. This was because the transpiration flux was the governing mechanism of the forest water balance in dry summer seasons.

Under conditions of meteorological drought, beech did not reduce transpiration rates and instead continued to exploit soil water reserves more extensively (by ~60 mm season$^{-1}$ on average). These differences resulted in higher summer drainage to deeper layers below the spruce canopy. During wet summers and all winter seasons, the soil was drier under spruce because the governing mechanism was interception, which was higher in the spruce canopy (by ~40 mm season$^{-1}$ on average). Hence, in these periods, the soil profile drainage was higher under the beech canopy.

The results suggest that with an increasing incidence of summer drought spells, soil water will be increasingly depleted through transpiration or soil evaporation in both forest types, with a greater effect in beech forests. The combined effects of climate and land cover change may thus increase the risk of soil drought and limit groundwater recharge. On the other hand, the increasing land cover by beech stands (accompanied by a shift in precipitation from the summer to the winter season) may result in increased winter drainage, as interception by leafless beech trees is lower than that of evergreen spruce.


### Code and data availability

Model code and all utilised data are available upon request.

### Author contribution

VS, MT, LV created the concept and set the methodology; NZ, VS, JT, JK carried the investigation; NZ implemented the model and created visualizations; NZ, VS and MB wrote the manuscript draft; all authors reviewed and edited the manuscript.

### Competing interests

The authors declare that they have no conflict of interest.

### Acknowledgement

This research was supported by the Czech Science Foundation (GA CR 24-10375S), the institutional support of the Czech Academy of Sciences, Czech Republic (RVO: 67985874, 67985939), and by the programme framework of the Strategy AV21. The authors would like to thank David Pěsta for conducting regular field measurements.



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
