# Peer review of "Divergent water balance trajectories under two dominant tree species in montane forest catchment shifting from energy- to water-limitation"

_Hydrology and Earth System Sciences, 2024_

## Referee Comment (RC1)

```
title: Review of "Future changes in water availability: Insights from a long-term
monitoring of soil moisture under two tree species" by Nikol Zelíková et al.
author: Reviewer
date: "2024-10-14"
MS No.: hess-2024-244
MS type: Research article
```

**General Comments**

(overall quality of the preprint)

**Scientific significance:**

The manuscript represent a substantial contribution to scientific progress within the scope of Hydrology and Earth System Sciences. It analyses long term soil moisture measurements from a spruce and a beech forest. The data were used to fit a water balance model to the sites. Based on the model results the authors investigate the sensibility of the water balance terms on short and long term changes in the environmental conditions, in particular precipitation (water and snow). Finally, they draw conclusions about the differences in the water balance of spruce and a beech forests. This topic is of great importance because the proportion of these two tree species in Europe is changing over large areas due to climate change.

**Scientific quality:**

The long term measurements are without doubt a big asset. In the comprehensible scientific approach, presumably all available measurements were used to calculate the terms oft he water balance. Nevertheless, the delineated approach includes mayor uncertainties that are not satisfyingly regarded. The measured quantities do not allow a direct partitioning of precipitation in evapotranspiration and groundwater recharge or runoff. Therefore, the partitioning is done by empirical models that can not clearly validated, i.e. the partitioning and the discussed individual water balance of the two forest stands is partly based on model assumptions and parameters that are derived on other sites.

Even so, I assume the authors used all measured data and information of the site and the results are convincing. So, the uncertainties just need to be more clearly highlighted and, if possible, quantified.

A weak point of the study is the approach used to calculate evapotranspiration. It does not explicitly regard the differences between spruce and beech. In my opinion the Penman-Montheith equation is state of the art and it allows the differentiation via canopy and aerodynamic resistance. A method to reduce the uncertainty are direct measurements of evaporation and transpiration. The authors already named sapflow measurements within the paper. Here I would like to point out that scaling to the forest stand is critical and that an underestimation of transpiration usually occurs.

Another uncertainty is the high spatial heterogeneity of soil moisture due to canopy and soil structure. The authors mentioned up to five measurement profiles. A description of the variability between the profiles with respect to canopy cover would help to establish confidence in the representativeness of these measurements.

As the authors stated, a big advantage of the long term measurements is the possibility to investigate trends in the time series (see line 63, henceforth the shorting L63 is used). However, I missed a discussion of whether or not changes can be observed over time. Subsection 3.1 shows the inter-annual variation of air temperature and precipitation but not of the soil water content and the other terms of the water balance. Also, "3.3 Climate-induced soil water regime and soil water fluxes" covers more seasonal changes at the site than changes induced by climate variability or change (long term changes).

As a distinct feature of the tree type specific water budget the authors discuss the inner-annual variation of the terms. It would be nice to have a visualisation of a typical annual cycle of soil moisture, evapotranspiration and drainage (something like a climograph).

**Presentation quality:**
In general, the scientific results and conclusions are presented in a well-structured way. The number and quality of figures/tables is adequate (except for Fig. 5, where the font sizes are too small for a printout).  The English is comprehensible and generally good, but there are some sentences that lack clarity and conciseness and need to be revised.

**Specific Comments**

In the following, I will switch to direct address to create a dialog.

As mentioned above the question arises: **Are the measurements representative for the sites?** Did you compare other measurements on the same patches? On L119, you write "One to four tensiometers were available for each measuring depth at each site, and the single value for a particular depth was taken as their average." First, what is meant by "single value", second could you illustrate the positioning of the sensors with respect to canopy cover, and third could you show the variability of the soil moisture for both sites?

Concerning the "**Vertical distribution of pressure heads**": Long term mean values over different seasons and conditions (Fig. 3) are difficult to interpret, as the differences between the measurement levels are small compared to the variability of the pressure head. I am wondering whether there is a significant deviation of the pressure head in a certain depth from the other levels, especially for beech. The categorisation according to precipitation is a good approach, however, Figure 4 shows that there is still a large degree of variation when considering a whole year. It would be interesting to see what the differences between levels and sites in the time domain look like (similar to flood statistics, i.e., what is the return interval of pressure heads below a certain value and how long do they persist).

In Figure 5 and the text, you use four soil moisture categories. Unfortunately, I couldn't find a clear definition.

L199: "Dry and wet years were identified by analysing the soil moisture regime in terms of the vertical distribution of pressure heads". Typical time series were given in Figure 5. Could you give a clear definition? Please explain the method or give a reference.

**Model calibration**: "The **entire period** of available data was used for model calibration". Validation of the model is therefore only partially possible at best. The given RMSE of the pressure heads are just an assessments of the quality of the fitting procedure (Btw: What method was used to optimise the parameters?). Usually, one part of the data is used to calibrate the model, and the other part is used for validation.

The model is calibrated with respect to the soil water content. The long-term means of the drainage fits well to the measured runoff. Although, it is assumed that beech and spruce stands experience the same drainage in the long term, that might not be realistic (see your discussion starting at L437). Could assess the error in S(t) and D(t)?

You write "However, the modelled high transpiration rates at the beech sites mostly follow from fitting to the high-resolution time series of measured local soil moisture data, which show lower values during the summer season compared to spruce, and simultaneous observations of no change in groundwater levels." (L440). However, this is no justification for the assumption that the ground water recharge below the beech is the same as below the spruce.

Please, explain (and discuss) the choice of your method for the **calculation of the evapotranspiration** ET. As far as I understood, you have all variables for the calculation of the actual ET available at your site. Why do you calculate the PET following a reduced approach (Oudin et al. 2005) and estimate the actual ET for the two sites from that?

What is the reason for calculating the Net longwave radiation (L155)? It seems not necessary, neither in your described model nor in the PET approach of Oudin et al. (2005). However, you cite Kofroňová et al. (2019), who used the Penman-Montheith equation to calculate the potential evapotranspiration (which is actually not correct, since the Penman-Montheith equation calculates the actual evapotranspiration). Which approach did you use Oudin et al. (2005), or the Penman-Montheith equation?

Concerning S(t) (L160): How is the influence of tree type regarded?

Looking for correlation between the terms of the water balance and environmental quantities (L326), why do you use the snow cover duration and not the precipitation during the winter season (water equivalent of the snow). I am not surprised by the weak correlation between snow cover duration and soil moisture, there can be long cold winters with snow cover but little precipitation and vice versa for warm winters. The usual argument, snow cover enhances infiltration, is not applicable at your site, as you wrote on L131: "surface runoff is not generated in the experimental catchment and all water directly infiltrates into the soil".

Results from literature and own observations get sometimes mixed up in the argumentation (see L382 ff. and L442: "The comparatively high transpiration rates of beech during the summer season were separately validated by measured sap flow (Brinkmann et al., 2016; Gebhardt et al., 2023)"). Please make clear what is your observation and what can you conclude from that, and finally compare it to literature.

**Technical Corrections**

(Minor errors and comments)

In the following I listed critical points (mostly about language) and gave some suggestions after the "==>".

**General points**

- At various points the text is not as precise as it should be for a scientific publication, please revise the manuscript carfully, e.g.
  - L20: "While long-term column averaged pressure head indicated drier soil at the spruce site overall, this was driven by the wettest years in the dataset."
  - L364: "The partitioning of the water fluxes in both stands was driven by different rates of interception (higher at the spruce site) and transpiration and soil evaporation (higher

under beech).":  The partitioning of the incoming precipitation is driven by vegetation type, soil conditions and other environmental conditions …

- o On L454 you conclude "This was because the transpiration flux was the governing mechanism of the forest water balance in dry summer seasons.", However, the transpiration was not measured, thus this could not be a conclusion, it is more an assumption.

- When listing quantities, please use "and" rather than a forward slash "/", as the latter is mathematically equivalent to division.

- Figures: writing of the units in squared brackets might be common but is wrong. The axis label represents the numbers. So I ask whether "pressure head [cm] = -800" is a correct statement? In comparison, "pressure head / cm = -800" or "pressure head in cm = -800" would be correct. I would recommend the later phrasing.

**Specific points**

L20: "While long-term column averaged pressure head indicated drier soil at the spruce site overall." ==> article missing or use plural, i.e. "the pressure head" or "pressure heads"

L22: "… drove complex but robust differences in flow partitioning between the forest types." ==> "… drove complex but robust differences between the forest types in regard to flow partitioning"

L25: "Estimated summer recharge" ==> "The estimated ground water recharge in summer"

L53: "greater"?: "Schume et al. (2004) and Šípek et al. (2020) reported greater soil profile drying during the vegetation season at beech sites." ==> "Schume et al. (2004) and Šípek et al. (2020) reported a stronger drying of the soil profile during the growing season at beech sites."

L60: "they provide only a partial view of the role of individual water fluxes" ==> "They only provide a partial insight into the role of individual water flows."

L123-125: Please clarify and shorten if possible

L135: Define $\Theta$ (it's probably the change in soil moisture in the topmost layer)

Eq. 7: $\Theta_{(t-1)}$ is not defined. Is it the measured $\Theta$?

L168: Please regard the unit conversion: $K_s$ is given in mm h$^{-1}$, whereas $D(t)$ is in mm d$^{-1}$

L188: "additional requirements" ==> "boundary conditions"

L200: check grammar

L232 ff.: "attained" ==> "reached"

L236: "vegetation seasons" ==> "vegetation periods" or "growing season"

L239: Please define the "four soil wetness categories" more precisely

L243 "the spruce site attained lower pressure head values than did the beech site" ==> "the spruce site reached lower pressure head values than the beech site"

Fig. 5: nice overview, but graphs are to small for printout, zooming reveals poor quality, it would be nice to see PET also

L270: To assess the RMSE values the reader should first be introduced to the values of the snow water balance.

L275: What do you mean with "SWBM efficiencies". I recommend deletion of "SWBM efficiencies in terms of the"

L278: "VWL" is not defined up to now.

"3.2.2 Simulated Water balance" contain several redundancies and could be shortened

L284: Please define "actual evapotranspiration (AET)".  Is it equal to $S(t) + P_{int}(t)$?

L285: "transpiration/soil evaporation (T+E)", Isn't that the same as S? (L134: "S(t) is the actual evapotranspiration rate (mm day−1)") ==> "transpiration plus soil evaporation S"

L287: "Beech reaches almost 100 mm higher T+E, and similarly, spruce reaches this level in the case of interception." ==> "The beech stand reaches almost 100 mm more S than the spruce stand, on the other hand, the evaporation from the interception storage in the spruce stand exceeds that of the beech stand to the same extent."

L348: "Seasonal precipitation also had a major influence on the differences in particular water fluxes between beech and spruce sites" ==> are there water fluxes between the beech and the spruce site? ;-)

L361: "measured pressure heads" ==> "measured soil water potentials"; The analysis also uses other measurements, doesn't it?

L429: "the tensiometer measurement limit" ==> "the tensiometer limit" or exactly "the measuring range of the tensiometer"

L430 "Eliminating this bias did not allow model fitting during dry periods." Yes, however, a model is usually not calibrated on the whole data set but only on a shorter calibration period and then applied to the whole data set.

L456: "These differences resulted in higher summer drainage to deeper layers below the spruce canopy." ==> These differences resulted in higher summer drainage to deeper layers below that of the spruce canopy.

---

## Referee Comment (RC2)

**Revision of Manuscript hess-2024-244**

**Future changes in water availability: Insights from a long-term monitoring of soil moisture under two tree species**

The authors evaluated the impact of climate forcings in the past 22 years (2000-2021) on the water balance components (evaporation, transpiration, drainage) in two experimental soil plots, one covered by Norway spruce, and the other one covered by European beech. The authors exploited a good data set of water potential data monitored by tensiometers installed at different soil depths. This wealth of data support a bucket model to simulate water balance.

The evaluation of this manuscript is based on the following questions:

1) Is it a novel work based on a reliable scientific technique?
2) Is it clearly structured and well-written?
3) Are the experimental design and analysis of data adequate and appropriate to the investigation?

The paper is well-written and well-structured. However, I would like to state that HESS is a high-ranked Journal and should receive novel, robust, scientifically-sound studies. My main concerns are:

1) The title mentions about future changes in water availability but the results refer to the past 22 years (2000-2021). I have no doubts that global warming is changing climate patterns in Central Europe. However, climate change can be detected only on very long time series by capturing decadal trends. In other words, climate change should be supported by data. The authors should state what is the baseline-historical climate regime in terms of rainfall and temperature observed in the past century. Climate change can be predicted by climate projections from 2020 up to 2100 which are based on scenarios depending on the carbon dioxide emissions (RCPs). There are many GCMs available in internet.
2) The authors present a detailed and interesting analysis on the impact of climate forcings on the water balance components and profile-average pressure head under two different land uses. However, what is the novelty of this article? I appreciate the unique long-term data set, but what is new if compared to the state of the art? How can readers exploit the findings of this study?

In the discussion session (line 395) the authors state that:

"This is the first study we know of to cover more than a few seasons and thus to provide a robust interannual comparison of soil moisture regime under beech and spruce canopies integrated over the entire soil column. Viewed at this scale, the comparison of soil moisture regimes proves to be

precipitation dependent. Our results have some clear implications for how combined changes to climate and forest species composition in Central Europe should be expected to impact forest water regimes. Climate change is expected to shift the intra-annual distribution of precipitation totals from summer to winter. We found that higher winter drainage under the beech canopy is even more pronounced with increasing winter precipitation. In combination with the increasing representation of beech trees in Central Europe, this can generally lead to higher winter groundwater recharge and runoff. By contrast, summer season differences indicate lower future recharge from beech forests given their higher transpiration demand, which generally consumes soil water even during drought periods. Overall, our results suggest that ongoing trends in climate and forest composition are pushing these forest systems from energy-limited towards water-limited states, at least on a seasonal basis."

It comes to no surprise that the comparison of soil moisture regimes proves to be precipitation dependent. The results related to this site-specific study (area of 1 km$^2$) cannot be representative for the impact of climate and land use change in Central Europe. The waster balance depends on soil depth, layering, and soil hydraulic properties, on the terrain features, on vegetation patterns and characteristics, on climate regimes and many other factors. The last sentence is usually supported by visualizing the Budyko curve which is used to understand the long-term balance between water availability and energy in a catchment (a region drained by a river or stream). It helps us analyze how much precipitation is evaporated versus how much becomes streamflow.

3) Another concern is on the use of a bucket model. Bucket models are usually used at coarse spatial scales where data are poor or inaccurate (regional to continental to global scales). The rich data set at plot scale in this study could support a Richards-based model which is more complex than the bucket model and provides a better performance in terms of model simulations.

4) Model calibration is poorly described. The authors used a local or global optimization tool? What's the objective function? The RMSE of what? Of pressure heads? Or else? The authors force the simulated annual cumulative drainage to be close to 360 mm year-1 because this value corresponds to the mean annual observed runoff. In this case the study area should be described more in detail by adding hydrogeological information to support this hypothesis which is strong. Then in the results, close to line 270 (please add continuous line numbers!), the authors mention about the model calibration against observed snow cover equivalents. In Line 274 the authors state that the calibration was done against observed soil water content that pop out of the blue. In M&Ms I do not see the description of soil water content sensors. I rather see only the installation of tensiometers. Did I miss anything?

5) The M&Ms would benefit from the use of a schematic figure that presents the overall study (measurements, modeling calibration/validation, data analysis, etc.)

---

## Author Comment (AC1)

[revised manuscript text omitted]

**Supplementary material**

**Table S1: Average soil hydraulic parameters of all soil layers derived from direct measurements. Each value is depicted by its mean ± standard deviation.**

| | Measured SHP | $\theta_r$ | $\Theta_s$ | $\alpha$ | $n$ |
|---|---|---|---|---|---|
| | 10 cm | 0.32 ± 0.03 | 0.70 ± 0.04 | 0.04 ± 0.01 | 2.10 ± 0.53 |
| Spruce | 35–45 cm | 0.18 ± 0.04 | 0.52 ± 0.03 | 0.04 ± 0.01 | 1.72 ± 0.15 |
| | 50 cm | 0.15 ± 0.02 | 0.48 ± 0.02 | 0.05 ± 0.01 | 1.58 ± 0.19 |
| | 70–75 cm | 0.15 ± 0.04 | 0.50 ± 0.04 | 0.07 ± 0.03 | 1.45 ± 0.13 |
| | 10 cm | 0.17 ± 0.02 | 0.53 ± 0.02 | 0.05 ± 0.01 | 1.36 ± 0.02 |
| Beech | 30 cm | 0.18 ± 0.01 | 0.49 ± 0.04 | 0.05 ± 0.01 | 1.55 ± 0.21 |
| | 45 cm | 0.17 ± 0.01 | 0.47 ± 0.01 | 0.05 ± 0.01 | 1.46 ± 0.01 |
| | 60 cm | 0.13 ± 0.02 | 0.42 ± 0.01 | 0.05 ± 0.02 | 1.48 ± 0.13 |

**Figure S2: A - Measured and modelled snow water equivalent (upper two panels) and B - soil water content (lower two panels)**

[Figure]

---

## Author Comment (AC2)

**Future changes in water availability: Insights from a long-term monitoring of soil moisture under two tree species**

**Nikol Zelikova et al.**

**Author´s response to Reviewer#1**

**Comment #1**

A weak point of the study is the approach used to calculate evapotranspiration. It does not explicitly regard the differences between spruce and beech. In my opinion the Penman-Montheith equation is state of the art and it allows the differentiation via canopy and aerodynamic resistance. A method to reduce the uncertainty are direct measurements of evaporation and transpiration. The authors already named sapflow measurements within the paper. Here I would like to point out that scaling to the forest stand is critical and that an underestimation of transpiration usually occurs.

Please, explain (and discuss) the choice of your method for the **calculation of the evapotranspiration** ET. As far as I understood, you have all variables for the calculation of the actual ET available at your site. Why do you calculate the PET following a reduced approach (Oudin et al. 2005) and estimate the actual ET for the two sites from that? Which approach did you use Oudin et al. (2005), or the Penman-Montheith equation?

**Response#1**

- Thank you for pointing out the difference in aerodynamic resistance between the two species, which we address below. Importantly, the model we used does differentiate between spruce and beech on two important processes: (1) the interception is estimated differently for both sites and the estimation is based on the measured characteristics, (2) soil water balance model parameters (namely theta_S, theta_R and Ksat) used for the estimation of beech and spruce transpiration and drainage are different as they were obtained by the model calibration on the different soil water regimes. This results in higher transpiration of beech in dry summer periods than for spruce – which is proven by the observed soil water regime.
- As the hydrological models are usually based on the two-stage modelling scheme – potential (PET) and actual evapotranspiration (AET) we have chosen Oudins approach for the estimation of PET.
- The reasons for the Oudin´s approach were:
  - o Oudin´s method represents a **robust approach** relying only on the air temperature, and therefore it can be used for the estimation of PET also in the periods with limited data availability without the loss of consistency in the input data series when they are replaced by the data from the neighbouring meteorological station
  - o **Side-experiments** utilizing the values of the P-M reference evapotranspiration aside from Oudins approach (in the period of available data) documented the influence of selected approach only on the soil water balance model parameters and not on the resulting water fluxes.
    - ▪ acceptable differences among P-M and Oudin's values of PET, especially when e.g. monthly means are considered. The day-to-day fluctuations are more averaged by Oudin (Fig.R1).
    - F

[Figure]

Fig. R1 Comparison of Penman-Monteith and Oudin PET

- the above-mentioned differences led to similar performance of the soil water balance model using both P-M and Oudin approach for the estimation of potential evapotranspiration (Fig. R2).

[Figure]

Fig. R2 Modelled soil moisture using Penman-Monteith and Oudin PET

- canopy specific parametrization of P-M approach by means of adjusted aerodynamic resistance (difference in canopy height was set to 5 m) resulted in the seasonal difference in PET of only 3.7 mm in 2010 and 1.5 mm in 2009. If the amount of soil water was changed by only 1%, it resulted in a change of AET by 9.4 mm in 2009 and 8.9 mm in 2010, respectively. This documents the limited influence of aerodynamic resistance compared to the influence of soil water availability (reflected in stomatal resistance) which is an inherent part of the model.
- **Literature review**: 1) Oudin et al (2005) paper showing reasonable results of hydrological models when using his approach compared to state-of-the art models in the conditions of limited data availability and 2) Touskova et al. (2025) paper showing a reasonable correspondence of Oudin's PET values and pan evaporation data in the Czech Republic and also with P-M reference evapotranspiration in terms of the seasonal sums
- The necessary **data for the Penman-Monteith (P-M) method are available only from 2008** and they originate from the nearby grass covered meteorological

station (300 m distance) at the forest opening. Hence, we do not have the opportunity to obtain site-specific information about wind profiles separately for beech and spruce forest.

**Comment#2**

What is the reason for calculating the Net longwave radiation (L155)? It seems not necessary, neither in your described model nor in the PET approach of Oudin et al. (2005). However, you cite Kofroňová et al. (2019), who used the Penman-Montheith equation to calculate the potential evapotranspiration (which is actually not correct, since the Penman-Montheith equation calculates the actual evapotranspiration).

**Response#2**

Yes, long-wave radiation was not necessary – it is a mistake in the manuscript text which will be deleted.

**Comment#3**

Another uncertainty is the high spatial heterogeneity of soil moisture due to canopy and soil structure. The authors mentioned up to five measurement profiles. A description of the variability between the profiles with respect to canopy cover would help to establish confidence in the representativeness of these measurements.

As mentioned above the question arises: **Are the measurements representative for the sites?** Did you compare other measurements on the same patches? On L119, you write "One to four tensiometers were available for each measuring depth at each site, and the single value for a particular depth was taken as their average." First, what is meant by "single value", second could you illustrate the positioning of the sensors with respect to canopy cover, and third could you show the variability of the soil moisture for both sites?

**Response#3**

We ensured the representativeness of our measurements for the sites in several ways.

- Our sites are even-aged, single species stands with closed canopies and no gaps; this relative homogeneity of vegetation makes representing bulk soil moisture/potential mostly a question of sufficient replication and avoiding placement of sensors at non-representative microsites.
- Locations of the plots in which the tensiometers were installed were carefully picked so that they will represent the **average slope** of the catchment and they will be located in **between trees**. Hence, they do not represent the places close to trees, which will be influenced by preferential water flow by stem flow, as well as they are not located in the forest openings. The measuring profiles are approximately 3.6 m from spruce trees and 2.7 m from beech trees. The average distance between two neighbouring trees is 5.4 m in spruce forest and 4.5 m in beech forest, indicating locating the measuring profiles approximately in between the trees.
- Both beech and spruce forests are of uniform age and the **spatial variability of canopy cover** represented by coefficient of variation of LAI is 12.8 % in spruce and 8.9 % in beech forest, respectively. The coefficient of **variation of soil moisture** ranged from 2.2-2.4 % for particular depths in beech and from 3.5 to 10.8 % in the spruce forest. The spatial variability of forest canopy and soil moisture measurements is of similar order as the error in precipitation measurement.
- In our previous work (Sipek et al., 2020), we have compared the average values of measured pressure heads for several depths in spruce site with another three profiles equipped with UMS T8 tensiometers located nearby (20 m from original profiles). The

comparison proved similar pressure head values demonstrating a **good correspondence with other measurements** (see Fig. R4).

[Figure]

Fig. R4 Comparison of pressure heads measured by UMS T8 (black lines) and Thies (red crosses) tensiometers at the SPR site in the period of 2009–2014. The grey area represents the malfunction of UMS T8 tensiometer and was not included in the statistical assessment. Each plot represents a particular depth of measurements (bottom left corner). RMSE stands for a root mean square error and R2 for a coefficient of determination (Sipek et al., 2020).

**Comment#4**

As the authors stated, a big advantage of the long term measurements is the possibility to investigate trends in the time series (see line 63, henceforth the shorting L63 is used). However, I missed a discussion of whether or not changes can be observed over time. Subsection 3.1 shows the inter-annual variation of air temperature and precipitation but not of the soil water content

and the other terms of the water balance. Also, "3.3 Climate-induced soil water regime and soil water fluxes" covers more seasonal changes at the site than changes induced by climate variability or change (long term changes).

As a distinct feature of the tree type specific water budget the authors discuss the inner-annual variation of the terms. It would be nice to have a visualisation of a typical annual cycle of soil moisture, evapotranspiration and drainage (something like a climograph).

**Response#4**

Thank you for the valuable comment. We have newly tested the existence of trends in soil moisture time series using a trend-free pre-whitening Mann-Kendall approach (Yue et al., 2002) and statistically significant negative trends were observed in both soil moisture time-series documenting gradual changes in soil water regime, which were also observed by the reported increasing occurrence of water limited seasons. This will be added to the manuscript.

A climograph is a thoughtful comment and the figure below (Fig. R5) documenting the differences between beech and spruce plots will be added to the manuscript.

[Figure]

Fig. R5 Average monthly sums of soil water balance components in beech and spruce forest

**Comment#5**

Concerning the "**Vertical distribution of pressure heads**": Long term mean values over different seasons and conditions (Fig. 3) are difficult to interpret, as the differences between the measurement levels are small compared to the variability of the pressure head. I am wondering whether there is a significant deviation of the pressure head in a certain depth from the other levels, especially for beech. The categorisation according to precipitation is a good approach, however, Figure 4 shows that there is still a large degree of variation when considering a whole year. It would be interesting to see what the differences between levels and sites in the time domain look like (similar to flood statistics, i.e., what is the return interval of pressure heads below a certain value and how long do they persist).

**Response#5**

Thank you for suggesting a better way of documenting depth differences in soil water regime between beech and spruce site. We will remove figures 3 and 4 and we will add the pressure head exceedance probability plot instead (Fig. R6). The manuscript text will be modified accordingly.

[Figure]

Figure R6. Exceedance probabilities of pressure head for particular depths for averaged the entire period (thick solid lines), dry year 2015 (short dashed lines) and wet year 2020 (long dashed lines). Green colour represents spruce and orange beech forest.

**Comment#6**

In Figure 5 and the text, you use four soil moisture categories. Unfortunately, I couldn't find a clear definition. L199: "Dry and wet years were identified by analysing the soil moisture regime in terms of the vertical distribution of pressure heads". Typical time series were given in Figure 5. Could you give a clear definition? Please explain the method or give a reference.

**Response#6**
The definition of the soil moisture is in the lines 245-254. It originates from the observed soil moisture regime:

- category A - spruce retained lower pressure heads throughout most of the season
- category B - only one single event when the beech site attained lower pressure heads than spruce
- category C - the pressure head decreased more pronouncedly at the beech site for a significant part of the summer season
- category D - refers to the seasons when the tensiometer measurement limit of – 865 cm was reached (mostly at the beech site)

**Comment#7**

**Model calibration**: "The **entire period** of available data was used for model calibration".

Validation of the model is therefore only partially possible at best. The given RMSE of the pressure heads are just an assessments of the quality of the fitting procedure (Btw: What method was used to optimise the parameters?). Usually, one part of the data is used to calibrate the model, and the other part is used for validation.

**Response#7**

Thank you for the valuable comment. We will add following important information about model validation, which was done prior to the overall model calibration presented in the manuscript. Our previous omission of this information unnecessarily undermines confidence in the results.

At the very beginning, we started with the standard procedure as we calibrated the model using several 5y calibration periods. For this purpose, we **split the period of interest into 4 sub-periods** – each covering 5y (2000-2004,2005-2009,2010-2014,2015-2019) and calibrated the model separately for each of these periods, always carefully maintaining the fit of drainage to the measured runoff. The model parameters were fit using the **genetic algorithm using the RMSE of volumetric water contents as an objective function**.

As the model parameters and also the model performance did not change substantially (see Fig. R7 below) we have chosen to calibrate the model for the entire period so that the water balance (i.e. discharge) can be maintained as close as possible to the measured long-term mean. The amount of drainage estimated from the water balance is more precise, and we could utilize this approach with only minor deterioration of an objective function compared to the situation when parameters from each of the 5y periods were used.

[Figure]

Fig. R7 Model performance when calibrated in particular periods. Values from first columns represent calibration from 2000 to 2004, the second and following columns represent the following calibration periods (2005-2009, 2010-2014, 2015-2019, and the last column is an overall calibration)

**Comment#8**
The model is calibrated with respect to the soil water content. The long-term means of the drainage fits well to the measured runoff. Although, it is assumed that beech and spruce stands experience the same drainage in the long term, that might not be realistic (see your discussion starting at L437). Could assess the error in S(t) and D(t)?

You write "However, the modelled high transpiration rates at the beech sites mostly follow from fitting to the high-resolution time series of measured local soil moisture data, which show lower

values during the summer season compared to spruce, and simultaneous observations of no change in groundwater levels." (L440). However, this is no justification for the assumption that the ground water recharge below the beech is the same as below the spruce.

**Response#8**

Thank you for pointing out this lack of clarity in the manuscript, which gives the impression that we assumed that the beech plot experienced the same drainage as spruce. In fact, we assumed similar or lower values of drainage based on the fact that in summer periods when no changes in groundwater level were observed, we observed more pronounced declines in measured soil water content under beech. This assumption is supported by the literature (most relevant papers cited), as studies on the topic predominantly report higher transpiration rates of beech.

Moreover, we have started the measurements of sap flow in August 2024 at seven trees at each site using Trunk Heat Balance method with EMS-81 sensors (EMS, Czech Republic). The monthly sums (August) of transpiration of 60.2 mm in the case of spruce forest and 76.2 mm in the beech forest were observed. Further, the soil water model run was extended to August 2024 and modelled monthly transpiration sums of 61.2 and 81.5 mm for spruce and beech forest were modelled. This indicates that the modelled differences in transpiration are in an acceptable agreement with measurements - although we are aware that one-month period is very short for a proper analysis and hence, we will not add this analysis to the manuscript.

To sum up, it arose from the facts that (1) the soil moisture declined more pronouncedly, (2) reported transpiration of beech is higher and (3) no changes in groundwater level were observed during these declines.

**Comment#9**

Concerning S(t) (L160): How is the influence of tree type regarded?

**Response#9**

The influence of tree type is reflected through different parametrization of the effective wetness (theta_E) restricting the rate of PET. The parameters include theta_S and theta_R, which govern the linear relationship of S(t) representing the rate of actual evapotranspiration to potential one based on the available soil moisture similarly to the approach of (Feddes and Rijtema, 1972). The different parameter values are documented in Table 1. In most cases, this results in the effective wetness ranging from 0.29 to 0.60 in the case of spruce and from 0.40 to 0.80 in the case of beech. Hence, in the case of beech plot, the rate of actual ET is following PET more closely.

We are aware that the utilized modelling approach is not describing the physiological behaviour of plants entirely, especially in the drought stress periods as more complex reaction to the water deficiency stress was reported both in the case of beech (Walthert et al., 2021) and spruce (Zweifel et al., 2002), but is is current state-of-the-art approach in hydrological modelling.

**Comment#10**

Looking for a correlation between the terms of the water balance and environmental quantities (L326), why do you use the snow cover duration and not the precipitation during the winter season (water equivalent of the snow). I am not surprised by the weak correlation between snow cover duration and soil moisture, there can be long cold winters with snow cover but little precipitation and vice versa for warm winters. The usual argument, snow cover enhances infiltration, is not applicable at your site, as you wrote on L131: "surface runoff is not generated in the experimental catchment and all water directly infiltrates into the soil".

**Response#10**

The information about snow cover duration was used to demonstrate the limited role of winter characteristics on the summer soil moisture (correlation coeff = 0.08). The same is valid for the maximum snow water equivalent (correlation coeff = 0.15), winter precipitation (correlation coeff = 0.09; see Fig. R8 using winter precipitation in comparison with original Figure R9) and the length of continuous snow cover (correlation coeff = 0.01). Both figures show similar limited influence of winter meteorological characteristics.

[Figure]

Fig. R8 Demonstration of winter precipitation (very left column) influence of summer soil moisture regime

[Figure]

Fig. R9 Demonstration of snow cover duration (very left column) influence of summer soil moisture regime

**Comment #11**

Results from literature and own observations get sometimes mixed up in the argumentation (see L382 ff. and L442: "The comparatively high transpiration rates of beech during the summer season were separately validated by measured sap flow (Brinkmann et al., 2016; Gebhardt et al., 2023)"). Please make clear what is your observation and what can you conclude from that, and finally compare it to literature.

**Response#11**

Yes, thank you for the notice. We did not want to mix up our results with the results from the literature. We will polish the mentioned parts of the text so it can be clearly distinguished what is the result and what are the comparisons with other authors.

**Comment#12**

Technical corrections

**Response#12**

Thank you for mentioning several inaccuracies in the text. We will carefully correct all points that you have mentioned.

References:

Oudin, L., Hervieu, F., Michel, C., Perrin, C., Andréassian, V., Anctil, F., and Loumagne, C. : Which potential evapotranspiration input for a lumped rainfall-runoff model? Part 2 - Towards a simple and efficient potential evapotranspiration model for rainfall-runoff modelling, J. Hydrol., 303, 290–306, doi:10.1016/j.jhydrol.2004.08.026, 2005.

Toušková, J., Falátková, K., and Šípek, V.: Estimating potential evapotranspiration in a temperate zone: The challenge of model selection, Water Res. Manag., 2024 (under review).

Šípek, V., Hnilica, J., Vlček, L., Hnilicová, S., and Tesař, M.: Influence of vegetation type and soil properties on soil water dynamics in the Šumava Mountains (Southern Bohemia), J. Hydrol., 582, 124285, doi:10.1016/j.jhydrol.2019.124285, 2020.

Yue, S., Pilon, P., Phinney, B., Cavadias, G.,:. The influence of autocorrelation on the ability to detect trend in hydrological series. Hydrol. Process., 16, 1807–1829.

---

## Author Comment (AC3)

**Future changes in water availability: Insights from a long-term monitoring of soil moisture under two tree species**

**Nikol Zelikova et al.**

**Author´s response to Reviewer#2**

**Comment#1**

The title mentions future changes in water availability but the results refer to the past 22 years (2000-2021). I have no doubts that global warming is changing climate patterns in Central Europe. However, climate change can be detected only on very long time series by capturing decadal trends. In other words, climate change should be supported by data. The authors should state what is the baseline-historical climate regime in terms of rainfall and temperature observed in the past century. Climate change can be predicted by climate projections from 2020 up to 2100 which are based on scenarios depending on the carbon dioxide emissions (RCPs). There are many GCMs available in internet.

**Response#1**

Thank you for pointing out this discrepancy in the current manuscript's framing of the study. We agree that it will be helpful for the reader to see a baseline at the appropriate scale to assess longer-term climate-driven shifts. We also agree the text needs to be clarified on this point: our intention was not to make specific projections of future water availability; rather, we examine how ecosystem functioning modulates the hydrological impacts of climate change. As relevant vegetation processes play out on timescales shorter than the overall climatic drivers (incl. differences in seasonal and sub-seasonal vegetation functioning), their effects on local water flux partitioning are readily apparent in our data. We will make this clearer both by changing the title and in the text itself.

To provide a baseline for the reader to evaluate climate-driven shifts in flux partitioning, we will add potential and actual evapotranspiration (PET and AET) and precipitation values since the start of local measurements in the experimental watershed in 1975, using the Budyko framework suggested by the reviewer (see Comment#3 below). The expected broader climate-induced shift is well supported by these local data (Fig. R10), which evidence an accelerating shift in the balance between atmospheric water supply and demand at a decadal time-scale.

[Figure]

Fig. R10 Ratios of actual and potential evapotranspiration to precipitation from the experimental watershed covering the period 1976 to 2020 shown within the Budyko curve reference frame; left: 5y averages, right: annual values.

We believe that adding this local time-series (1976-2020) will be sufficient to situate our study within the broader climatically driven pattern. It shows our data cover a period (2000-2021) in which the climatic drivers are forcing a gradual shift from a fully energy-limited state (1976-2000) through more co-limited regimes, in which the most recent outlier years already show strict water-limitation. We do not feel our work needs to belabour the fact of climate change or its expected broader-scale, exogenous impacts on the hydrologic balance any further – there are plenty of studies demonstrating these, sufficiently referenced in the manuscript (lines 38-45).

Our work specifically poses questions about how climatic drivers interact with vegetation processes to produce hydrologic flux partitioning and ecosystem function. The processes we studied occur over sub-annual time-scales and their effects can be adequately evaluated in inter-annual differences in site/watershed hydrologic balance. The core 22-year dataset is thus entirely sufficient to evaluate their effects. It is also by far the longest available of its kind that we are aware of.

We see little added value from the use of GCMs to produce climate projections for our study. Mainly, this is because we are not aiming to make any projections. Instead, we analyse observable ongoing changes to advance process understanding of climate-vegetation interactions beyond what is currently represented in GCMs. Using GCMs that cannot adequately represent the feedbacks between climate, vegetation, and soil hydraulic properties to make projections of future water availability strikes us as a highly uncertain means of addressing our questions. Rather, we believe process understanding needs to be improved before GCM projections can meaningfully encompass such feedbacks. In response to this comment, we will make our aims clearer in the manuscript introduction (around lines 63-67).

**Comment#2**

The authors present a detailed and interesting analysis on the impact of climate forcings on the water balance components and profile-average pressure head under two different land uses. However, what is the novelty of this article? I appreciate the unique long-term data set, but what is new if compared to the state of the art? How can readers exploit the findings of this study?

**Response#2**
We appreciate these questions from the reviewer as they indicate that the manuscript still needs to state the novelty, significance, and usefulness of our work more clearly. The state of the art is currently a mosaic of short-term studies with mutually contradictory results (Manuscript lines 52-55, 388-392). Their unresolved contradictions stem mostly from the limited duration of each study, which makes each partial result context-dependent on the specific climatic conditions of the study period (lines 56-63).

The main novel advantage of our long-term dataset is the ability to make **robust interannual comparisons** over a range of climate conditions (Line 395). This enabled us to disentangle interactions of specific climatic (summer/winter precip, temperature) and vegetation (phenology, rooting, hydraulics) drivers of forest hydrologic response to dry vs wet years and seasons. A second advantage of the dataset is **depth coverage over the rooting zone** and high replication, allowing us to overcome site-scale heterogeneity and study limitations due to limited vertical extent of measurements (e.g., lines 390-392). Finally, the integration of this dataset within long-term observations from the experimental watershed enables us to impose **closure on the hydraulic balance** and estimate individual fluxes for both forest types.

Our main novel contribution, enabled by this unique combination of dataset advantages, lies in showing which climatic variables have driven water limitation so far (atmospheric water supply

more so than demand), and which vegetation processes most exacerbate or dampen it in the studied forest types. We showed which vegetation traits or processes become important to the hydrological balance under which conditions, including during previously unobserved water-limited years. The revealed interactions produce feedbacks that will ultimately lead to differences in function and fate between these important forest types. Specific novel findings with broader significance include:

- whether beech or spruce forest soils end up drier in the growing season depends on intra-annual precipitation distribution due to seasonal differences in flux partitioning by the two forest types (lines 366-375); this helps to resolve previous contradictions in the literature as a majority of studies are limited to a single number of years.
- differences in winter drainage between the forest types increase with winter precipitation (lines 400-402); under expected summer to winter shifts in precipitation, this novel interaction should enhance beech forests' water limitation and affect their role in baseflow generation and intra-annual storage/discharge timing, with implications for forest and water management.
- beech hydraulic function capable of sustaining transpiration during drought interacts with summer atmospheric water balance (PET/P) to exacerbate recharge reductions in warmer or drier years (lines 374-382).
- by contrast, higher spruce contributions to summer drainage due to reduced transpiration initially persist as water limitation begins to affect the system, but eventually decline to zero under water *supply* limitation (lines 386-387).
- collectively, these findings demonstrate divergent patterns of forest hydraulic functioning under water limitation in summer/winter, due to atmospheric supply/demand, which advance process understanding in support of model development or direct application to ecosystem and water management.
- the difference in soil moisture between the forest types is dominated by depths > 30cm (lines 392-394); this is not entirely unexpected due to known rooting depth differences, but quantifying this dominance remains a finding valuable for scientific practice when the vast majority of soil moisture measurements are done at depths < 20cm.

A final point of novelty is the recent occurrence of annual-scale water limitation of AET, which is unprecedented over the 40+ year instrumented period and entirely unexpected in this montane system. The entire range, including our sites, is classically thought of as energy-limited, not just under "baseline" (1961-1990) climate but for past millennia. Our results include some of the first observations of differences in hydrologic functioning of these cold, humid montane forest types under water limitation. Anticipated climatic trends make the publication of these findings all the more timely.

We believe diverse readers will make use of our findings because cross-scale interactions in land-atmosphere feedbacks such as these are one of the key sources of uncertainty in predicting shifts in ecosystem function and composition driven by climate change. Our study contributes towards filling key gaps in the required process understanding by empirically resolving specific processes that contribute to differences in forest hydrologic functioning under shifting climate. For researchers, this understanding contributes towards the next generation of models and projections. Practitioners can use it to evaluate interactions between the ecosystem and water management. We will reorganise the discussion to improve the explanation of the novelty and significance of the results.

**Comment#3**

It comes to no surprise that the comparison of soil moisture regimes proves to be precipitation dependent. The results related to this site-specific study (area of 1 km2) cannot be representative for the impact of climate and land use change in Central Europe. The water balance depends on soil depth, layering, and soil hydraulic properties, on the terrain features, on vegetation patterns and characteristics, on climate regimes and many other factors. The last sentence is usually supported by visualizing the Budyko curve, which is used to understand the long-term balance between water availability and energy in a catchment (a region drained by a river or stream). It helps us analyze how much precipitation is evaporated versus how much becomes streamflow.

**Response#3**

Thank you for suggesting the Budyko conceptual framework. We agree this is a very productive framing for our work and we will add a figure to illustrate this statement. Fig. R10 (above) shows clearly that both study sites have slowly transitioned from energy-limitation towards co-limitation over recent decades and in the driest recent years, they indeed switch to a clearly water-limited regime.

Importantly, the plot also shows increasing divergence between the hydrologic functioning of the sites with increased water limitation. This further underlines the importance of not only the climate but also its interactions with vegetation to the hydrologic balance, placing the processes our study examines in context. We will **update the discussion** to make use of this framing in explaining the significance of our findings.

We agree with the reviewer that our study does not achieve representativeness at the landscape, let alone regional scale. We appreciate this comment as it gives us an opportunity to better clarify the significance of our work despite this lack of representativeness.

The landscape-scale implications of our work do not depend on the watershed's broad representativeness so much as its particular landscape position. Due to the region's geography, montane forests in headwater catchments represent the areas of high precipitation and low evapotranspiration. Through both locally higher inputs and intra-annual storage, forested montane headwater catchments play an outsize role in baseflow generation, supporting regional hydrological stability. The broader landscape's (i.e. downstream) water regimes will thus be particularly sensitive to their seasonal functioning under climate change.

Due to land use patterns, these montane forests also represent the majority of strictly protected areas (IUCN categories Ia, Ib, and II) in Central Europe, with a dominant proportion of our two tree species. This includes the twinned Bohemian/Bavarian Forest national parks directly adjacent to our sites. Understanding the ecohydrology of montane forests dominated by these species will be one of the keys to regional biodiversity and ecological conservation during the ongoing hydroclimatic shift.

In sum, understanding vegetation-mediated land-atmosphere feedbacks in montane, mid-slope beech and spruce forest such as these is particularly important to projections of future ecological and hydrological dynamics across the region.

In terms of direct generalisation, our study is most like a paired watershed study in hydrology or a common garden experiment in vegetation ecology. Our study design allows the key processes to be examined at the appropriate scale, without needing to represent the entire landscape. The resulting process understanding is always only transferable to an extent circumscribed by an adequate consideration of conditions in the study system. Scaling the effects of the processes we described to an entire landscape would require a separate exercise that would take into account the factors mentioned by the reviewer, but is beyond the scope of our study.

That said, some of our novel findings generalise directly. Our catchment lies close to the cold, humid end of the spectrum of Central European climate zones (e.g., unit C7 on the Quitta Climatic Classification, Vondrakova et al., 2013 https://doi.org/10.1080/17445647.2013.800827). Observation of an annual-scale flip to a water-limited regime here is not only entirely out of line with historical experience. It is also strongly indicative for large parts of the generally warmer, drier Central European landscape: if it has started to happen here, it will be happening at least episodically in most places.

We will improve the discussion section to clarify both the limits on generalisation of our conclusions and their broader landscape significance despite these.

**Comment#4**

Another concern is on the use of a bucket model. Bucket models are usually used at coarse spatial scales where data are poor or inaccurate (regional to continental to global scales). The rich data set at plot scale in this study could support a Richards-based model which is more complex than the bucket model and provides a better performance in terms of model simulations.

**Response#4**

Solving the Richards equation for soil flows was our choice in our previous study (Sipek et al., 2020) dealing with a 5y data set at the same site. We found that modelling the soil water regime this way at multiple depths continuously for 5 years introduces significant uncertainty. RMSE ranged from 99 to 176 cm for the column average pressure head (it was even larger when we assessed specific depths), parameters of the root water uptake function needed to be far from physically reasonable values, and parameters of the soil water retention curves also had to be adjusted from their measured values.
The reasons for the doubtful performance were namely:
- large amounts of rock fragments in the soil. If a certain percentage of the profile is formed by the rock fragments, then the vegetation will extract more water from the areas between those rock fragments to fulfil the water demand. This could result in a higher actual drop in observed pressure heads, which would not be represented in the model.
- the hydrophobicity of the soils may result in non-uniform drainage of water into deeper soil layers and formation of a shallow biomat flow. The percolation of water can then be limited only to certain locations (eventually bypassing the measurement probes).
- occurrence of preferential flow in the forested catchment can cause non-sequential reaction of soil moisture sensors at different depths
- soil hydraulic properties are described by the soil water retention curves, which strictly define properties of the porous media. However, if 21 years are modelled continuously, soil properties undergo gradual changes, which were not measured.

Hence, for this long-term study we have chosen simpler bucket type of the model as (1) it is sufficient to answer questions posed (soil column water balance) without adding more complexity, (2) it uses "Feddes" type of equation for the estimation of plant water use as Richards-based models, (3) it is more convenient for the simulation of longer periods, (4) the soil column is represented by one unified domain with column average soil hydraulic properties, which is beneficial especially when the soil encompasses a lot of rock fragments.

**Comment#5**

Model calibration is poorly described. The authors used a local or global optimization tool? What's the objective function? The RMSE of what? Of pressure heads? Or else?

Then in the results, close to line 270 (please add continuous line numbers!), the authors mention about the model calibration against observed snow cover equivalents. In Line 274 the authors state that the calibration was done against observed soil water content that pop out of the blue. In M&Ms I do not see the description of soil water content sensors. I rather see only the installation of tensiometers. Did I miss anything?

**Response#5**

Thank you for the valuable comment. We will clarify important information about model calibration in the manuscript. The type of line numbering is pre-described by the journal template for manuscript submission.

The model parameters were fit using the **genetic algorithm using the RMSE as an objective function**. The model was calibrated in two steps (lines 171-179). First, parameters of the degree-day snow accumulation/melt model were calibrated using measured snow water equivalents. Second, the remaining model parameters were calibrated using a measured soil water regime represented by pressure heads. For soil water balance modelling, the measured pressure heads were used to calculate the volumetric soil water content by means of the van Genuchten (1980) function. The function parameters were retrieved from the measured retention curves specific for each site and depth (lines 181-184).

**Comment#6**

The authors force the simulated annual cumulative drainage to be close to 360 mm year-1 because this value corresponds to the mean annual observed runoff. In this case, the study area should be described more in detail by adding hydrogeological information to support this hypothesis which is strong.

**Response#6**

Yes, this is a fundamental part of the modelling framework. It is based on the previous hydrogeological survey which documented crystalline bedrock in the catchment which only allows water circulation in the weathered zone and does not communicate with adjacent catchments. Therefore, the hydrological catchment corresponds well to the hydrogeological catchment (Hrkal et al., 2009). (lines 85-87) and the assumption, while strong, is also well supported.. We will make this clearer in the manuscript.

**Comment#7**

The M&Ms would benefit from the use of a schematic figure that presents the overall study (measurements, modeling calibration/validation, data analysis, etc.)

**Response#7**

Thank you for the comment, we will add the required figure into the manuscript or supplements.

[Figure]

Fig. R11 Scheme representing the workflow of the study

References:

Šípek, V., Hnilica, J., Vlček, L., Hnilicová, S., and Tesař, M.: Influence of vegetation type and soil properties on soil water dynamics in the Šumava Mountains (Southern Bohemia), J. Hydrol., 582, 124285, doi:10.1016/j.jhydrol.2019.124285, 2020.

---

## Referee Report (RR1)

title: Review of "Divergent water balance trajectories under two dominant tree species in montane forest catchment shifting from energy- to water-limitation" by Nikol Zelíková et al.

subtitle: Former title "Future changes in water availability:

Insights from a long-term monitoring of soil moisture under two

tree species"

author: Reviewer
date: "2025-07-09"
MS No.: hess-2024-244

MS type: Research article

**Comments on the 2th manuscript**

(overall quality of the preprint)

**Scientific significance:**

Please refer to the first review from 2024-10-14.

**Scientific quality:**

The authors provided a thoroughly revised manuscript, where even the title has changed.

The title and also the discussion indicates the investigation of the water balances of two forest stands. The within the text and also the section title in the methods part (2.3) this is restricted to a "soil water balance model".

Due to the lack of data, the used water balance model is based on very simple classical approaches. Nevertheless, the long time series of measured soil water content are valuable and need to be investigated. The analysis and discussion is encompassing, but needs at some points clarification.

**Specific Comments**

The following comments are also made largely with reference to the first review.

**Comment#1**

of the first review was "A weak point of the study is the approach used to calculate evapotranspiration. It does not explicitly regard the differences between spruce and beech. ..."

This Problem is still not solved. The differences between the sites is regarded by the soil conditions only. As the potential evapotranspiration for both sides is obviously calculated following a simple temperature function by McGuinness and Bordne (1972).

$$PET=rac{Re}{\lambda
ho}rac{Ta+5}{100}$$
 for  $Ta+5>0$  , otherwise  $PET=0$

The actual evapotranspiration (without interception) is then derived just by multiplying PET with the effective soil water content as proposed by (Feddes and Rijtema, 1972).

There seems to be no consideration of the difference in the phenological phases between the beech stand and the spruce stand and their influence on the transpiration.

If one compares the main four environmental drivers for tree transpiration (stomatal conductance, see Stewart, 1988): radiation, water vapour pressure deficit, temperature and soil moisture deficit. In mid In European forests, the influence of soil moisture deficit on tree transpiration is often the least significant. However this might change during droughts.

Since data for the Penman-Monteith (P-M) method has only been available at this location since 2008, it might be reasonable to extend the time series with this simple approach. However a comparison with a more refined method should be shown in the manuscript or at least in the supplementary.

**Comment#3**

The relative homogeneity of vegetation is not a strong indicator for the representativity of the soil moisture measurements. The convenient placement of the sensors is also only a necessary condition but not sufficient. Soil structure, particularly in soils with a high skeleton content, is much more important for water drainage, root penetration and finally plant-available water.

The additional measurements with the UMS T8 indicate a reasonable correlation with the Thies sensors. Please, put the graphics also in the supplement.

**Comment#4**

I still miss a graphical presentation of the soil moisture changes observed over time. Could you add the average values of soil moisture and runoff to Figure 2.

**Comment#5**

Thank you for the exceedance probabilities of pressure head (Fig. 3). I am wondering why the probability for the entire period does not run in between the dry and the wet years. Why is the pressure head by 100 % probability of exceedance not the same for entire period and for wet years? Likewise, why is the pressure head by 0 % probability of exceedance not the same for entire period and for dry years? This should be the minimal and maximal value contained in both datasets respectively.

**Comment #6**

Please, place a concise description of the categories directly after L293 "... seasonal development of their measured pressure heads.", i.e. move the part after L314 up to L293 and complete it with values (L314 is not a sum up, but a qualitative definition of the categories).

**Comment #7**

The description of the model parameterisation and validation has improved with subsection 2.4, but it still needs some clarification. Please, make short sentences and use tables or lists for description of parameters. Clearly indicate which parameters are calibrated using which data and quality criteria.

If you use the runoff for calibration you need to define the areal distribution of beech and spruce in the catchment.

L230 You state "4 sub-periods for cross-validation". Cross-validation is a statistical technique used to evaluate how well the results of a model or analysis will generalize to an independent, unseen dataset. It involves partitioning the available data into subsets, training the model on some of these subsets, and testing it on the remaining ones. This process is repeated multiple times to ensure that the model's performance is robust and

not just tailored to a specific portion of the data.... It is not clear what is the unseen dataset which you use for the cross-validation.

I would assume that there is a difference in the parameter set for summer and winter, at least at the spruce site.

L240: "forward modelling"? Did you start with a known model and predict observations? It is more "inverse modelling", where you derive the model parameter, i.e. the model, from observations.

L241 "Besides the model run in the period of available soil water potential measurements (2000–2021), the model was run also from 1975 to 1999 using the calibrated model parameters and available air temperature and precipitation sums to quantify annual AET for the entire observation period." This description is not really reproducible.

**Comment #9**

How is the influence of tree type regarded?

You respond "The influence of tree type is reflected through different parametrization of the effective wetness (theta\_E) restricting the rate of PET." However, the parameters theta S and theta R depend only on the soil type and soil structure, not on the tree type.

**Comment #10**

These are indeed a complex relationships; one could also investigate the dependence on the soil moisture of the previous year or the runoff.

**Comment #1b**

Fig. 7: Budyko plots are interesting. Please, compare the slidgly changed form in Renner et al. (2014). in my opinion it is somewhat clearer.

It is also to consider that in this study PET is only a linear function of the air temperature. A trend in PET is therefore initially a trend in temperature. AET is then the relative filled soil water storage times "Temperature" plus interception.

For PET/P < 1 the system is just energy limited, crossing the line of PET/P = 1 the system is additionally water limited. Instead of Fig 5 a) with the 5 year sums I would prefer a time series of AET/PET it might be that the shift between the two spaces in time is better visible using this relation.

**References**

Renner M, Brust K, Schwärzel K, Volk M, Bernhofer C (2014) Separating the effects of changes in land cover and climate: a hydro-meteorological analysis of the past 60 yr in Saxony, Germany. Hydrol. Earth Syst. Sci. 18:389–405

Stewart JB (1988) Modelling surface conductance of pine forest. Agricultural and Forest Meteorology 43:19–35

**Presentation quality:**

In general, the scientific results and conclusions are presented in a well-structured way. The number and quality of figures/tables is adequate (apart from the font sizes, they are often too small to print out). The English is comprehensible and generally good, but there are still some sentences that lack clarity and conciseness and need to be revised. Please, make the sentences as short as possible.

**Technical Corrections**

The PDF file contains a few minor errors and comments.

**Divergent water balance trajectories under two dominant tree species in montane forest catchment shifting from energy- to water-limitation**

- Nikol Zelíková1,2, Jitka Toušková1, Jiří Kocum1,3, Lukáš Vlček1, Miroslav Tesař1, Martin Bouda4,5, Václav Šípek1

  1 Institute of Hydrodynamics of the Czech Academy of Sciences, Pod Patankou 30/5, Prague, 160 00, Czech Republic
- Department of Water Resources and Environmental Modelling, Faculty of Environmental Sciences, Czech University of Life Sciences Prague, Kamýcká 129, Praha-Suchdol, 165 00, Czech Republic
   Department of Physical Geography and Geoecology, Faculty of Science, Charles University in Prague, Albertov 6, Prague,
  - 120 00, Czech Republic

    4 Department of Plant Ecophysiology, University of Hohenheim, Garbenstraße 30, Stuttgart 70599, Germany
  - 5 Institute of Botany of the Czech Academy of Sciences, Zámek 1, Průhonice, 252 43, Czech Republic

Correspondence to: Václav Šípek (sipek@ih.cas.cz)

**Abstract.**

15

surface. Therefore, both climate and land cover changes impact water resource availability. This study aimed to determine the differential effects of climate change on the soil water regime of two common Central European montane forest types:

Vegetation interacts with both soil moisture and atmospheric conditions, contributing to water flow partitioning at the land

- Norway spruce (Picea abies L.) and European beech (Fagus sylvatica L.). A unique dataset, including 22 years (2000–2021) of measured soil water potentials, was used with a bucket-type soil water balance model to investigate differences in
- evapotranspiration and groundwater recharge both between the forest types and across years. Results revealed an accelerating transition from a fully energy-limited state towards water-limitation, with evidence of strict water-limitation in recent outlier years, unprecedented in this system. While long-term column-averaged pressure heads indicated drier soil at
- the spruce site overall, this was driven by the wettest years in the dataset. Seasonal and interannual variability of meteorological conditions drove complex but robust differences between the flow partitioning of the two forest types, which diverged further with increasing water-limitation. Higher snow interception by spruce (27 mm season-1) resulted in drier soil
- below the spruce canopy in the cold season. Higher transpiration by beech (70 mm season-1) led to increasingly drier soils over the warm seasons causing lower ground water recharge (25 mm season-1). Low summer precipitation inputs exacerbated soil drying under beech as compared to spruce. These suggest that expected trends in regional climate and forest species

composition may interact to produce a disproportionate shift of recharge from the summer to the winter season.

**1 Introduction**

35

65

| 33 | temants crusive. A major obstacle to advancing process understanding is the lack of long-term observations of variables with                                                                   |
|----|------------------------------------------------------------------------------------------------------------------------------------------------------------------------------------------------|
|    | direct mechanistic relevance, such as water potential (or hydraulic head). Water potential in soil and plants suffers from a                                                                   |
|    | noted information gap despite being key to our understanding of land-atmosphere interactions (Novick et al. 2022). Soil                                                                        |
|    | moisture status integrates the fluxes of the entire hydrological cycle and in turn exerts significant control over key Earth                                                                   |
|    | system processes (Legates et al., 2011; Humphrey et al. 2021). As water potentials directly drive the soil-plant-atmosphere                                                                    |
| 40 | water flows that are tightly coupled with other land-atmosphere fluxes, addressing $\underline{\text{this}}$ gap offers a promising pathway to                                                 |
|    | resolving major uncertainties in ecosystem fate and functioning (Trugman et al. 2018, Green et al., 2019) during the                                                                           |
|    | transition to previously unobserved hydroclimatic regimes. After centuries of relative climatic stability (Brázdil et al., 2022),                                                              |
|    | a clear rise in average and maximum air temperatures has been affecting Central Europe since the last part of the 20th                                                                         |
|    | century (Zahradníček et al., 2020). Increased air temperature has induced higher atmospheric water demand contributing to                                                                      |
| 45 | the severity of recent droughts (Možný et al., 2020). Although, long-term annual precipitation sums have not changed in the                                                                    |
|    | past (Brázdil et al., 2021) and are not expected to change significantly in near future (Svoboda et al., 2017), the occurrence                                                                 |
|    | of seasonal precipitation deficits causing severe soil drought is projected to increase (Hari et al., 2020). Increased water                                                     |
|    | demand combined with seasonally reduced water supply is expected to shift the region from energy- toward water-limitation                                                                      |
|    | of evapotranspiration over the coming decades (Denissen et al., 2022).                                                                                                                         |
| 50 | One of the less well understood consequences of ongoing climatic changes is a shift in forest species composition, which has                                                            |
|    | the potential to further affect water fluxes in the soil-plant-atmosphere system (Maxwell et al., 2018). The two most frequent                                                          |
|    | tree species in central European forests are beech (Fagus sylvatica L.) and spruce (Picea abies L.). As spruce thrives in                                                                      |
|    | colder and moisture-rich conditions, its stands are increasingly being replaced by beech (Daněk et al., 2019). This climate-                                                                   |
|    | $induced\ transformation\ of\ \underline{montane}\ forests\ \underline{has\ potential\ implications\ for\ ecosystem\ ecohydrological\ function}.\ Each\ of\ these$                             |
| 55 | $species\ has\ distinctive\ physiological\ and\ architectural\ properties\ \underline{such\ as\ leaf\ morphology}\ and\ \underline{phenology},\ \underline{rooting\ depth}\ (\underline{Jost}$ |
|    | et al., 2012), xylem structure and function (Tyree & Zimmermann, 2002), or stomatal control during dry periods (Gebhardt                                                                       |
|    | et al., 2023). Their specific ecohydrological characteristics and strategies may not only determine their fates under                                                                          |
|    | hydroclimatic change but also yield divergent effects on the water balance through contrasting rates of interception                                                                           |
|    | (Savenije, 2004), soil water fluxes, water storage dynamics, and thus soil water regimes (Schume et al., 2004).                                                                                |
| 60 | At present, available studies comparing soil moisture regimes under these common tree species provide ambiguous results                                                                        |
|    | due to their limited duration. Schume et al. (2004) and Šípek et al. (2020) reported a stronger drying of the soil profile during                                                              |
|    | the growing season at beech sites. By contrast, Schwärzel et al. (2009), Rötzer et al. (2017), and Kuželková et al. (2024)                                                                     |
|    | observed greater soil drying under spruce than under beech. Some of these differences may partly be explained by                                                                 |
|    | contrasting soil hydraulic properties at the sites compared. The main limitation shared by such studies; however, is their                                                                     |

Making ecohydrological predictions in a non-stationary state of the Earth system requires detailed process understanding that remains elusive. A major obstacle to advancing process understanding is the lack of long-term observations of variables with

limited temporal extent. The periods of the observation range from one day (e.g., Jost et al., 2012) to several years (Schume

fluxes. They are limited by the variability of climatic conditions during the study period. Moreover, short-term studies 70 cannot capture long-term changes in the characteristics of droughts, such as higher temperatures (Groissord et al., 2021) and flash droughts (Qing et al., 2022) and therefore their second-order effects via the given species. Hence, the availability of a long-term data series is crucial not only to observe trends, but also as a tool to better understand processes and natural

et al., 2004; Schwärzel et al., 2009; Zucco et al., 2014; Korres et al., 2015; Huang et al., 2016; Rötzer et al., 2017). The longest periods of analyses so far lasted from 4 to 5 years (Wang et al., 2018; Šípek et al., 2020; Gebhardt et al., 2023). The results of short-term studies are difficult to interpret as they provide only a partial insight into the role of individual water

variability in a period of changing climate and land cover (Huntingford et al., 2014; Milly et al. 2015). This study aims to advance process understanding by disentangling the effects of climate and forest composition on water 75 fluxes as these ecosystems transition from energy- to water-limitation. We focused on the impact of two forest types, monospecific Norway spruce and European beech, on the soil water regime in an experimental montane catchment in Bohemian Forest, Czechia. The study benefits from a unique 22-year-long dataset of measured soil water potential in the two forest types that enables us to make robust interannual comparisons for the first time. Long-term observations of the experimental catchment allow us to impose closure on the hydraulic balance to estimate individual fluxes and to compare the current evapotranspiration regime with previous decades. Together with its depth coverage over the rooting zone in each 80

85 process-based soil water balance model, and (3) determine the main climate dependency of the soil water regime under both tree species. 2 Data and Methods

stand, these advantages allow the present dataset to yield comprehensive insight into the studied forests' ecohydrological function during the ongoing hydroclimatic transition. To reveal how climatic drivers interact with vegetation processes to produce hydrologic flux partitioning, we: (1) analyse seasonal differences in measured soil water potential between the two forest types, (2) estimate the soil water balance components (evapotranspiration and drainage) at the two sites using a

The study is based on extensive field measurements of soil moisture regime and necessary hydrometeorological variables in a Central European montane catchment including spruce and beech covered sites. The water balance of both sites was

estimated using the bucket type soil water balance model. The workflow of the study is presented in Supplementary material

The Liz experimental catchment, Czechia (49°04'N, 13°41'E) (Fig. 1), served as the experimental area for this study. It is located in the Bohemian Forest on the border between Czechia and Germany. The catchment area is approximately 1 km2. Its

elevation ranges from 828 m a. s. l. (at the outlet) to 1,070 m a. s. l. It is located in the cold region (unit C7 of the Quitt Climatic Classification, Vondrakova et al., 2013) of an otherwise humid continental climate (unit Dfb of the Köppen

**(Fig. S1).**

90

95

**2.1 Study site**

beech outlet

Climatic Classification (Tolasz et al., 2007 according to Köppen, 1936). During the study period 2000–2021 (and the preceding period of catchment measurements, 1975–1999) it had an average annual air temperature of 7.2 (6.4) °C and an average annual precipitation of approximately 847 (842) mm. The monthly average maximum temperature is 16.5 (15.5) °C

in July, and the minimum is -1.9 (-2.3) °C in January. More precipitation arrives during the May-October growing season than the rest of the year: 515.7 (471.2) mm compared with 331.9 (370.9) mm, respectively. Mean annual snow cover duration is 133 (147) days. The annual potential evapotranspiration (PET) determined by the air temperature-based method (Oudin et al., 2005) is 560.7 (521) mm. The annual runoff height from the catchment is approximately 352 (317) mm,

100

105

110

representing ~40% of the total precipitation.

Prague meteorological Liz site station BvC beech

Figure 1: Overview of the experimental site (© CUZK 2024) and soil profiles (© Přemysl Fiala).

Crystalline bedrock in the catchment only allows water circulation in the weathered zone and does not communicate with adjacent catchments, such that the hydrological catchment corresponds fully to the hydrogeological catchment (Hrkal et al., 2009). This observation underpins a fundamental assumption of our modelling framework: that all water from precipitation

approximately 30 m: the spruce site is located at 855-860 m a.s.l., and the beech site is located at 885-890 m a. s. l., both with a slope of 7.5° and an eastern aspect. Both spruce and beech canopies tend to suppress understory vegetation, which was accordingly absent at both sites (Fig. 1). The leaf area index (LAI) was measured throughout the 2022 season on a 120 monthly basis and showed a seasonally stable value with an average of 3.7±0.5 in the spruce site and seasonally variable values in beech ranging from 1.1±0.2 at the beginning and end of growing season (May and September) to 4.7±0.5 in the middle of the growing season. A visual inspection of the root depth distribution (when excavating the soil) revealed that the

generates measurable runoff at the gauging station, which is well supported by the hydrogeological survey. The majority of the area is covered by nearly pure spruce forest, with a dominance of 120–140-year-old Norway spruce (*Picea abies L.*) (> 85% of the canopy cover). In several places, the spruce forest is penetrated by 100–120-year-old beech stands (Fagus

Two experimental sites within the Liz experimental catchment were chosen for this study: one with Norway spruce (Picea abies L.) and the other with European beech (Fagus sylvatica L.). The elevation difference between the two sites is

site and 80.2%-18.1%-1.7% at the beech-covered site. The soil water permeability is relatively high ranging from 518 cm.day-1 at the bottom of the soil profile to 1700 cm.day-1 in the topsoil horizon. The humus A horizon (0–10 cm), together with surface organic horizon O (5–10 cm thick at beech stand and 10–15 cm at spruce stand), is followed by a Bvs/v horizon (down to 50 cm at beech site and to 30 cm at spruce site) and finally by a BvC horizon with a significant amount of larger 130 than sandy particles (>50%). Both soil profiles are presented in Fig. 1.

The soil at both sites can be classified as moderately deep loamy sand dystric Cambisol (IUSS, 2015), with an average soil depth of approximately 100 cm. The percentages of sand-silt-clay fractions are 73.2%-24.2%-2.6% at the spruce-covered

sylvatica L.).

115

125

**2.2 Field measurements The meteorological variables used in this study were air temperature (Fiedler RV12/RK5, Czech Republic) and precipitation**

roots were present only in the upper 40 cm of the spruce site and down to 100 cm of the beech site.

(Meteoservis MRW 500, Czech Republic), which were measured at 10-minute intervals during the entire twenty-two-year period (2000-2021). The meteorological station is located circa 400 meters away from two experimental plots outside the 135 forest. Moreover, the experimental catchment is instrumented with discharge and groundwater level measurements.

Discharge was also measured at the 10-min time step, and the groundwater level was recorded manually every week

throughout the entire investigated period. Average daily air temperatures, precipitation sums and discharges were collected from 1975. The snow water equivalent (SWE) was measured manually three times per week since 2000. Soil water potential

data were acquired from permanently installed soil tensiometers (Adolf Thies GmbH, Germany) measuring pressure heads at five depths (15, 30, 45, 60 and 90 cm). Soil water potentials were recorded manually three times a week during the growing

ranging from 0 cm to -865 cm (-85 kPa). Up to four tensiometers were available for each measuring depth at each site over the entire measurement period (at least 2 replicates 93 % of the time), and we used their average for a particular depth as the

season (mid-May to mid-October) from 2000 to 2021. The measuring range of these tensiometers included pressure heads

points representing average site slope and distance between the trees (3.6/2.7 m from tree in spruce/beech forest when the average distance in between two adjacent trees is 5.4/4.5 m). This resulted in same order of spatial variability of LAI (coefficient of variation was 12.8/8.9 % for spruce and beech) and soil moisture (coefficient of variation was 2.3/6.3 % for spruce and beech) in both forests and good correspondence of soil water potentials with another three profiles equipped with 150 UMS T8 tensiometers located nearby (Sipek et al., 2020). The average soil column pressure head was estimated as a

weighted mean of five soil layers (each represented by one measurement depth). The soil profile was considered to have a uniform depth of 100 cm. The measured pressure heads were used to determine differences in soil water regimes between the stands, as they better demonstrated the stands' behavioural differences during dry conditions, which were of interest to the

2012), the HBV model (Seibert and Vis, 2012) or the VIC model (Liang et al., 1994)). The modification for this study is

 $\frac{d\theta(t)}{dt} = P_{TF}(t) - S(t) - D(t)$

site-representative value. Given the fully closed, even, monospecific canopies at our sites, the representativeness of the measurements was ensured by avoiding placing sensors at micro-sites subject to preferential flows. We used measurement

study. 155

145

2.3 Soil water balance model

The conceptual model used in this study was a modified form of the soil water balance model (SWBM), developed by

Brocca et al. (2008, 2014). The bucket-type of the model was used as (1) it is sufficient to answer questions posed (soil column water balance) without adding more complexity, (2) it uses "Feddes" type of equation for the estimation of plant

water use as Richards-based models, (3) it is more convenient for the simulation of longer periods, (4) the soil column is represented by one unified domain with column average soil hydraulic properties, which is beneficial especially when the

160 soil encompasses a lot of rock fragments. Moreover, several widely used hydrological models use similar bucket/reservoir modelling approaches for the determination of soil water regimes (e.g., the Soil Water Assessment Tool (Arnold et al.,

based on the replacement of the infiltration parameter (the Green-Ampt equation) by throughfall  $(P_{TF})$ , as surface runoff is not generated in the experimental catchment and all water directly infiltrates into the soil. Therefore, the following soil water 165 balance Eq. (1) was used:

where  $\theta(t)$  is the average volumetric water content at a day (t),  $P_{TF}(t)$  is the throughfall (mm day-1), S(t) is the actual

 $P_{TF}(t) = P_{OAR}(t) - P_{INT}(t)$ 170

interception capacity from every single precipitation event. The interception capacity of 2.2 mm was derived by Kofroňová et al. (2021) for the same experimental site. In the case of beech stands, the summer interception capacity was calculated

where  $P_{OAR}$  represents the measured open area precipitation (mm day-1) and  $P_{INT}$  is the estimated interception (mm day-1) for a given location. Spruce interception in the summer season (May to October) was estimated based on the deduction of the

(1)

(2)

6

evapotranspiration rate (mm day-1) and D(t) is the drainage rate (mm day-1). The Eq. (2) for  $P_{TF}(t)$  is given as:

where *a* is an empirical coefficient (-) and *b* is the soil cover fraction (=LAI/3.0) (-). Daily values of LAI were acquired from linear interpolation between monthly measured values (May–September) conducted by a LI-COR 2000 Plant Analyser in 2022 (Toušková et al., unpublished results). The calibration of *a* parameter was performed so that the fraction of intercepted precipitation was allowed to range between 15 and 20%, which is an ordinary interception loss of beech canopies (Gerrits et al., 2010). For the winter season (November to April), linear regression functions linking open area snow water equivalent to

(LAI):

200

that below the forest canopy were used (Šípek and Tesař, 2014). The regression equations are based on the measured snow water equivalents in the forest openings and below the spruce (Eq. 4) and beech (Eq. 5) canopies for a period of ten years and are in the form:

using a general formula by von Hoyningen-Hüne (1983) and Braden (1985) applying seasonal variation in the leaf area index

 $P_{INT} = a \cdot LAI \left( 1 - \frac{1}{1 + \frac{b \cdot P_{OAR}}{a \cdot lAI}} \right)$

(3)

and are in the form:  $SWE_{TF}(t) = SWE_{OAR}(t) \cdot 0.595 \tag{4}$   $SWE_{TF}(t) = SWE_{OAR}(t) \cdot 0.679 \tag{5}$  where  $SWE_{OAR}$  is the snow water equivalent (mm day-1) in the open area and  $SWE_{TF}$  is the snow water equivalent under the

forest canopy (mm day-1).

Potential evapotranspiration (PET) was estimated using the Oudin et al. (2005) approach, which offers reliable estimates of PET for long-term water balance studies in the Central European region (Toušková et al., 2025). This approach provided a

consistent PET estimate based on data available for the entire observation period (1975–2021), whereas data needed for more sophisticated approaches are not available for the first decades. The actual evapotranspiration (*AET*) was found as the sum of *P*INT and soil evapotranspiration rate *S* (comprising soil evaporation and plant transpiration) was then estimated based on the linear decrease in its potential rate with decreasing effective soil water content as proposed by (Feddes and Rijtema,

on the linear decrease in its potential rate with decreasing effective soil water content as proposed by (Feddes and Rijtema, 1972) according to the following Eq. (6) and Eq. (7):  $S(t) = PET(t) \cdot \Theta_e$  (6)

$$S(t) = \underbrace{PET(t) \cdot \Theta_e}$$

$$\Theta_e = \left[ \frac{\Theta_{(t-1)} - \Theta_r}{\Theta_s - \Theta_r} \right]$$
(6)

contents (mm), respectively and  $\theta_{(t-1)}$  is modelled volumetric water content at previous day (t-1). The drainage component D(t) is a nonlinear function of  $\Theta_e$ :

where PET is the potential evapotranspiration (mm day-1) and  $\Theta_{exs}$  are the effective, residual and saturated soil water

 $D(t) = K_{\rm S} \Theta_e^{3 + \frac{2}{\lambda}}$

(8)

where  $K_s$  is the saturated hydraulic conductivity (mm day-1) and  $\lambda$  is the pore size distribution index (-) linked to the textural structure of the soil layer, which was set to 0.5. In this case, the flow is assumed to be gravity driven, with drainage consisting of deep percolation.

the experimental catchment lies in an area with regular snow cover. The degree-day method (Gupta, 2001) was chosen for 210 this purpose because it has been proven to be efficient in the Central Europe (Girons Lopez et al., 2020).

The original SWBM does not include a snow module; hence, snow accumulation and snowmelt had to be considered first, as

**2.4 Model parameterisation, validation, and forward simulation**

The model was calibrated with the genetic algorithm in two separate steps: one focused on the additional snow module and

the second on the original SWBM parameters using fixed values of snow parameters from the first step. In each case, the RMSE of the model response variable (snow water equivalent and soil water content, respectively) was used as the objective

215

function. The calibrated parameters of the snow module were the snowfall correction factor (SFCF), two threshold air temperatures—

one for snow to occur  $(T_{snow})$  and the second for the snowmelt to begin  $(T_{melt})$ —and the degree-day factor controlling the rate of snowmelt based on the air difference between the average daily air temperature and the threshold temperature (DDF). These were calibrated separately for each winter season so that the input for the soil water model was as accurate as possible.

220 The remaining model parameters were calibrated against the soil water content at both the beech and spruce sites. The calibrated parameters of SWBM were saturated ( $\theta_s$ ) and residual ( $\theta_r$ ) soil volumetric water content and saturated hydraulic

conductivity  $(K_s)$ . To obtain soil water content for calibration, the measured pressure heads were used to calculate the volumetric soil water content by means of the van Genuchten (1980) function. The function parameters were retrieved from 225

the measured retention curves specific for each site and depth (see Table S2 in Supplementary material). For more information about the determination of the soil water retention curves, we refer to Šípek et al. (2020). In addition to the minimisation of the RMSE, the model was calibrated with two boundary conditions: (1) simulated drainage from both sites must be approximately 360 mm y-1, which is a value obtained from the long-term measured runoff from the area, with the

beech site constrained to values equal or lower than the spruce site in accordance with our observations and higher 230 transpiration of beech (Brinkmann et al., 2016, Gebhardt et al., 2023) and (2) beech summer interception loss will be within 15–20% of the open area rainfall, which corresponds to the range reported by Gerrits et al. (2010).

(2000-2004, 2005-2009, 2010-2014, 2015-2019) and calibrated the model separately for each of these periods. In each

To evaluate model fit, we first split the period of interest into 4 sub-periods for cross-validation, each covering 5 years

case, we constrained drainage to fit the measured runoff. Model error in cross-validation was on the same order as measurement error (max. RMSE

Figure 2: Average air temperatures (upper two panels) and precipitation sums (bottom panel). The red columns represent the summer seasons (May-October), and the blue columns represent the winter seasons (November-April). Dashed lines represent season average values.

Pressure-head values were higher at the beech site, with a long-term median of -155 cm compared to -255 cm for

**3.1.1 Vertical distribution of pressure heads**

260

265

270

275

the spruce site. However, despite the higher median pressure-head values recorded at the beech site, the occurrence of low pressure-heads was more frequent here as reflected by higher exceedance of pressure values lower than -400 cm from the depth of 30 cm and deeper (Fig. 3). Differences in the vertical distribution of pressure heads were visible, namely, in the topsoil layer (depth of 0–15 cm), where soil under spruce reached permanently lower pressure head values than that under beech. The overall depth distribution of the pressure heads was more uniform under spruce — documented with flatter slope of curve describing the exceedance of pressure heads in all depths (Fig. 3). In contrast, the pressure head depth distribution under beech trees exhibited greater propensity to drying, especially in the bottom soil layers. The slope of exceedance curve is steeper namely between pressure heads of –200 cm and –400 cm. As the soil gets drier then the soil water potential is lower under beech. The beech site, despite having higher pressure heads on average, was therefore more susceptible to more

Intensive drying than the spruce site.

The differences between the beech and spruce site were less pronounced during the wet years (e.g., year 2020 represented by long-dashed lines in Fig. 3) but the soil under beech was noticeably drier in dry years (see example dry year 2015 represented by short-dashed lines in Fig. 3). Although the differences among the sites were small in wet years, lower pressure heads were observed at the spruce site at all depths. In contrast, during the dry year of 2015, the soil under spruce site was wetter (reached a higher column average median pressure head) than at the beech site even in the top soil layer (depth of down to 15 cm). Below the depth of 45 cm the pressure head of -850 cm was exceeded in more than 50 % of

records under beach and only up to 10 % in the case of soil under spruce canopy. Hence, the differences in pressure heads might be even greater, as the tensiometer data reached their limit more frequently at the beech site than at the spruce site;

thus, even lower pressure heads were likely to occur at the beech site. If the number of dry years increase in the future, the soil under beech will therefore become drier during the growing seasons.

Figure 3: Exceedance probabilities of pressure head for particular depths for averaged the entire period (thick solid lines), dry season 2015 (short dashed lines) and wet season 2020 (long dashed lines). Green colour represents spruce and orange beech forest.

**3.1.2 Soil wetness trend and categories**

285

290

We found significant negative trends in both daily soil moisture time-series, 0.7 mm yr-1 in beech (p-value 0.001), 0.2 mm yr-1 in spruce (p-value 0.0015), documenting gradual changes in soil water regime which correspond to the increasing occurrence of water limited seasons. Looking closer, we divided years into four soil wetness categories based on the typical

two sites. At the beginning of every summer season (May), the spruce site reached lower pressure head values than did the beech site (the average difference in pressure heads was 130 cm). Typically, as the season progresses, the pressure heads at the beech site decrease more than those at the spruce site. However, this was not valid for the wet seasons of 2002, 2005, 2020, and 2021, when spruce retained lower pressure heads throughout most of the season (see Fig. 4), as no precipitation deficit was observed (category A). For those seasons, the difference between the two sites was negligible, with their average

295

beech site)

seasonal development of their measured pressure heads. The evolution of average pressure heads for each month of the summer season over the measured period (2000–2021) is depicted in Fig. 4. At both sites, a similar pattern of decreasing

pressure heads from the onset of the summer season can be observed. However, there are noticeable differences between the

values fluctuating between -100 and -200 cm. In the other few years, when above average precipitation seasonal sums were <a href="reached">reached</a> (category B, including the years 2006, 2010, 2012, 2014, and 2016), there was only one single event when the beech site <a href="reached">reached</a> lower pressure heads (below -400 cm), which was usually ended by rainfall higher than 50 mm·day-1. In contrast, in the periods with below average precipitation, the pressure head decreased more pronouncedly at the beech site for a significant part of the summer season (category C included, e.g., years 2007, 2012 or 2019, as shown in Fig. 4). With even more prominent precipitation deficits (in 2003, 2008, 2015 and 2017), the beech site was the first and often only site to reach the tensiometer measurement limit of -865 cm (category D) – up to ten times more often than the spruce site.

reach the tensiometer measurement limit of -865 cm (category D) – up to ten times more often than the spruce site, especially in the bottom soil layers. Real pressure head values were likely significantly lower. As lower pressure heads cannot be recorded at the beech site with tensiometer measurements (measuring limit was reached) and pressure heads at the spruce site only seldom approached this limit, the differences between both sites were higher than documented by sensors. The effect on our analysis was likely insignificant as the implied differences in the amounts of water retained would be rather small. By the end of the season, pressure head values slowly increased, with beech still maintaining lower pressure

head values than spruce.

To sum up, we have used four categories of soil moisture regime for further analysis:

• category A - spruce retained lower pressure heads throughout most of the season

• category B - only one single event when the beech site attained lower pressure heads than spruce

category B - only one single event when the beech site attained lower pressure heads than spruce
 category C - the pressure head decreased more pronouncedly at the beech site for a significant part of the summer season
 category D - refers to the seasons when the tensiometer measurement limit of -865 cm was reached (mostly at the

Figure 4: Daily precipitation (P) (black columns) and soil column average pressure heads at beech (orange line) and spruce (green line) sites in all investigated years divided into four wetness categories (A-D) defined by pressure head values. The red dashed line represents the pressure head of -400 cm used for the division of categories A and B.

3.2.1 Model calibration result The modified SWBM model was used to obtain evapotranspiration and drainage fluxes over a period of twenty-two years

3.2 Modelling of evapotranspiration and drainage

**(2000–2021) at both spruce and beech sites. The mean RMSE values (2000–2021) for the snow module were 7.1 mm (beech) and 9.5 mm (spruce), which are in accordance with Sípek and Tesař (2017), who modelled snow cover dynamics**

325

330

Table 1. Calibrated soil water balance model parameters

453.0

0.50

depicted in Supplementary material (Fig. S4b).

precipitation fitted to the measured SWE is shown in Fig. S4a. The resulting mean RMSE (2000–2021) were 2.5% and 2.8% for the spruce and beech sites, respectively. The modelled long-term drainage was 353 mm year-1 for beech and 365 mm year-1 for spruce. The average annual discharge for the

from 2009 to 2014 and reached an RMSE value of 9.1 mm in a spruce stand. An example of the modelled cumulative snow

experimental Liz catchment was 360 mm, which was very close to the modelled values. The final parameters of the SWBM  $(\theta_{s}, \theta_{r}, K_{s}, \lambda)$  for each site are documented in Table 1. Examples of modelled and observed volumetric water contents are

λ  $K_s$ **RMSE**  $\theta_s$  $\theta_r$ Spruce 514.4 79.8 165.9 0.50 2.5 %

0.0

Beech

340

3.2.2 Simulated Water balance

The total actual evapotranspiration (AET; encompassing transpiration and soil evaporation (S) and interception ( $P_{int}$ ) and

21.0

0.50

2.8 %

drainage attain similar values at both plots on average. The total AET is approximately 540 mm season-1, and the drainage is

between 350 and 360 mm season-1 (Table 2). The beech reaches almost 100 mm more S than the spruce stand, on the other hand, the evaporation from the interception storage in the spruce stand exceeds that of the beech stand to the same extent.

The resulting AET values therefore do not differ greatly from each other because S and interception tend to compensate for

345

each other between stands, which is hence also reflected in similar drainage.

Even though the winter seasons are characterised by lower precipitation sums than the summer seasons (approximately 1/3

of the annual precipitation), the spruce forest had, on average, a higher rate of interception (133 mm season-1) due to

defoliated beech forest (106 mm season-1) (Table 2). However, from the AET perspective, the difference in interception is partially alleviated by slightly higher transpiration and soil evaporation under the beech canopy at the beginning and end of

350 the winter season (14 mm season-1). Nevertheless, the interception rate and winter transpiration at the spruce site resulted in a lower amount of water available for infiltration and therefore a lower modelled soil water content during the winter

14

months. The drier soil in spruce forests regularly represents an initial condition for the summer season. A higher soil water

soil evaporation from the soil column. AET stands for actual evapotranspiration. 2000-2021 **DRY 2015** WET 2020 Winter season Summer season SPR SPR SPR SPR **BEE** BEE SPR **BEE BEE**

mm season-1 on average).

355

content below the beech canopy was a reason for higher modelled drainage during the winter season at the beech site (by 12

Table 2: Modelled soil water balance components (mm) at the spruce and beech sites. S represents transpiration and

BE

282

103

385

901 923 499 340 Precipitation 561 S 261 345 270 357 221 297 46 62 213 Interception 275 204 270 194 201 139 133 106 143 **AET** 536 549 540 551 422 436 179 168 356

352 324 145 161 Drainage 365 352 162 174

205 180 360

In the summer, transpiration flux significantly affected the water balance at both sites-as, it was noticeably higher in the beech forest (see Table 2). The interception pattern of both stands was preserved, with spruce having higher interception (142 mm season-1) than beech (103 mm season-1). The differences in the soil water content were therefore caused by the transpiration in the beech stands (by 70 mm season-1). Hence, soil under spruce trees retained (with the ongoing summer 365 season) more water than soil under beech trees, where soil moisture was more effectively used for higher transpiration of beech trees, especially during dry spells. The wetter soil under spruce (in the majority of summer seasons) resulted in higher

drainage by 25 mm season-1 on average.

3.3 Interannual comparison of climatic drivers of seasonal soil water regime and soil water fluxes

Figure 5 shows the relative rankings of individual study years according to snow cover duration, air temperature, summer 370 precipitation (May-October), and their classification into the four wetness categories according to the resulting pressure head dynamics, shown in Fig. 4. The dominant factor controlling the soil water regime in the growing season was the amount of summer precipitation. A

significant soil moisture deficit could develop even following a winter with abundant snow. Fig. 5 clearly shows the direct link between pressure head and summer precipitation, where lower pressure heads are linked mainly to years with lower

Pressure head in cm 2020 2017 2005 2021 2019 2013 2004 2010 **2015** 2009

seasonal precipitation, and higher pressure heads are linked to years with abundant precipitation. The correlation coefficient between summer precipitation and soil moisture regime category was 0.80 (significant at 0.05 probability level). Two marginal categories (A and D) were always linked to specific climatic conditions (see Fig. 5). Category A, denoting wet soil (hence small differences between beech and spruce sites), was always determined by above average precipitation amounts and below average air temperatures observed in the summer season. Category D, representing the very dry soil moisture

regime, was always accompanied by low observed precipitation amounts in the summer season. Two middle categories (wetter B and drier C) tend to be connected primarily with above (in the case of B) and below (category C) average precipitation sums. The influence of preceding winter snow cover and summer season air temperatures was ambiguous, as seen in the frequently strongly mismatched placement of particular seasons along these axes in Fig. 5, compared to the resulting soil wetness category. The correlation coefficient with soil moisture regime were 0.30 and 0.08 for summer air

temperature and snow cover duration, respectively. Higher correlation coefficient was also observed for the summer vapour

Summer season (May-October)

-900

pressure deficit (VPD) attaining the value of 0.61 (not shown in the Fig. 5).

14.5

Winter season

(Nov-Apr)

2001 2007

2000 2014

375

380

385

390

2006 0 150 12.0 855 Snow cover duration in days Air temperature in °C Precipitation in mm Soil moisture regime

category (A-D), as shown in Fig. 4. The most pronounced deviations from the observed link between summer precipitation sums and the soil moisture regime

Figure 5: Average air temperature and precipitation sums for each summer season (represented by one horizontal line) encompassing the preceding winter snow cover duration. Each season is ultimately linked to a specific wetness

were in the 2013 and 2007 seasons, with above average precipitation but a drier soil moisture regime. This was caused by a

temperature (Fig. 2), and by the extreme floods in 2013, when the catchment received 1/3 of all summer precipitation in June but saw below average precipitation amounts during the rest of the season. These two factors caused a drier soil moisture regime even when above average precipitation sums were recorded. These results therefore document how different rainfall conditions influence the development of soil moisture content and the different behaviours of beech and spruce in growing season (Fig. 4).

near absence of snow cover observed in the winter of 2006/2007, accompanied by the highest recorded winter air

Figure 6: Differences between spruce and beech modelled soil water fluxes (AET, D) during summer (May to October, orange colour) and winter (November to April, blue colour) in relation to precipitation. AET can also be divided into INT and S (upper panel).

405

Seasonal precipitation also had a major influence on the differences between beech and spruce sites in particular water fluxes. (Fig. 6). Differences in all fluxes could be positively or negatively related to seasonal precipitation sums with the exception of winter transpiration and soil evaporation (S). The differences in winter and summer interception, winter actual

contrast, summer transpiration and soil evaporation, summer actual evapotranspiration (governed by transpiration) and winter drainage were negatively related to precipitation sums. The largest absolute differences in water fluxes between the stands were recorded during wet summer seasons. The most pronounced discrepancies were in the rates of transpiration and soil evaporation (higher for beech plots; up to 80 mm season-1), summer interception (higher for spruce plots; up to 55 mm season-1) and drainage (higher for spruce plots; up to 45 mm season-1). The lowest differences occurred during the dry winter seasons. The differences in the winter seasons were generally less prominent, usually below 40 mm season-1.

evapotranspiration (governed mainly by interception) and summer drainage increased with increasing precipitation. By

4 Discussion

420

**4.1 Transition from energy and water limitation The studied catchment falls within a montane system classically thought of as energy-limited not just under "baseline"**

period. Incipient water limitation at the annual-scale was first observed for the drought year 2003 and four times since, with entirely unprecedented examples of outright water-limitation over the years 2015 and 2018. Our dataset thus offers some of 425 the first observations of the hydrologic functioning of these previously cold and humid montane forest types under water limitation. a) b) 1,2 1,2 Water limit Water limit 1,0 1,0 1996-2000 1986-19902006 0,8 0,8 2011-2015 1991-1995 2015 2018

(1961–1990) climate but over previous millennia (Schafstall et al., 2024). Our results show gradual soil drying following an

accelerating shift in the balance between atmospheric water supply and demand (Fig 7). The transition from energy- toward water-limitation predicted for the coming decades (Denissen et al., 2022) is in fact already apparent over our measurement

Evaporative ratio (AET/P) Evaporative ratio (AET/P) 2016-2020 0,6 0,6 2011 2001-2005 20b7 0,4 0,4 PET/P < 1 PET/P < 1 PET/P > 1PET/P > 1 0,2 0,2 energy-limited water-limited energy-limited water-limited 0 0 0 0,2 0,4 0,6 8,0 1,0 1,2 0 0,2 0,4 0,6 8,0 1,0 1,2 1,4 Aridity index (PET/P) Aridity index (PET/P) Figure 7. Ratios of actual and potential evapotranspiration to precipitation from the experimental watershed covering the

period 1975 to 2020 shown within the Budyko curve reference frame – (a) 5-year averages and (b) annual values. Green points represent the spruce site and orange points beech.

435 the seasonal precipitation sums, even though observed trends in the catchment over the period 1975–2021 show that significantly increasing annual atmospheric demand (PET) (slope 1.6 mm y-1, p-value 1.91E-06) rather than insignificant changes in precipitation supply (P) drives increased aridity over the long term, the differences in flux partitioning in the driest years were strongly dependent on the seasonal precipitation sums (Figs. 5-6). With increasing water limitation, the

seasonal precipitation patterns and vegetation processes will become increasingly important drivers.

trend of atmospheric demand will cease to exert direct control over the water balance (P-AET), while interactions between

comparisons of the soil water regime between dry and wet years (Fig. 8), allowing modelled soil water fluxes under beech and spruce canopy to reveal the interactions between forest cover, climate, and soil moisture. Differences in winter soil

moisture regime were determined mainly by the higher interception of the spruce canopy, which resulted in higher pressure heads under beech causing more drainage compared to the spruce site. When the precipitation in the following summer

We found that increased water limitation enhances differences in annual evaporative ratio between beech and spruce forest, indicating divergence of their water balance in a drier climate. The differences in soil moisture were strongly dependent on

4.2 Vegetation and climate interactions in the soil moisture regime

150

100

40

**Our unique 22-year long dataset of measured soil water potentials, air temperatures and precipitation sums enabled robust**

440

445

450

40

season was high, only minor differences in pressure heads were recorded between stands, even though the spruce site maintained slightly lower pressure heads throughout most of the season (as a winter season legacy effect). The resulting differences remained small as the higher interception of spruce did not exceed the higher rate of transpiration of beech in the growing season. **Increasing** water deficit a) Wet year b) 2000-2021 c) Dry year 100 100 100 300 Drier soil Soil water spruce and drainage in mm 80 Drier soil **Drier soil** 250 250 under spruce 200 200 60 60

Ly Figure 8. A monthly water budget under spruce and beech canopy in wet (2002) and dry (2015) year and its overall averages

150

100

50

40

20

150

As the growing season advanced, transpiration became an increasingly important factor in the soil moisture regime. The balance between interception and transpiration and soil evaporation resulted in greater drainage under the spruce canopy. In 455

seasons with prominent precipitation deficits (Fig. 8c), the soil at the beech site consistently dried out more than at the spruce

|     | soil volume and water in its root zone, especially at greater depths (Čermák et al., 1995; Schwärzel et al., 2009; Gebauer et           |
|-----|-----------------------------------------------------------------------------------------------------------------------------------------|
|     | al., 2012). Beech also has a greater tissue-specific hydraulic conductance due to favourable anatomical and morphological               |
| 460 | traits, allowing it to supply leaves with water more efficiently at a given root-zone water potential (Tyree & Zimmermann,              |
|     | 2002). As a result, beech behaves more anisohydrically, maintaining transpiration rates in the face of drier soils, in contrast         |
|     | to the more isohydric spruce, whose lower ability to supply water to its foliage requires it to restrict transpiration earlier as       |
|     | the soil dries out (Čermák et al., 1995; Zweifel et al., 2002; Schume et al., 2004; Hochberg et al., 2017; Gebhardt et al.,             |
|     | 2023). Schwärzel et al. (2009) and Floriancic et al. (2022) reported higher evaporation from soil and litter under beech stands         |
| 465 | compared to spruce. Additional factors possibly affecting differences in soil water regimes include lateral flow, which is              |
| I   | reportedly more common at beech sites (Jost et al., 2012), and root water redistribution (Burgess et al., 1998). In dry                 |
|     | summers, the drainage remained higher under spruce canopy, although the difference between the stands decreased as the                  |
|     | difference between interception and transpiration declined.                                                                             |
| 470 | Robust interannual comparisons of the soil moisture regime under beech and spruce canopies integrated over the entire soil              |
|     | column allow this study to resolve the contradictory results of previous work limited in scope of over space or time. While             |
| I   | Schume et al. (2004) and Šípek et al. (2020) observed drier soil during the growing season under a beech canopy, Schwärzel              |
|     | et al. (2009) found the opposite. In the latter case the more prominent drying under spruce was attributed to the nonuniform            |
|     | and rocky soil compared to beech site. Rötzer et al. (2017) and Kuželková et al. (2024) also reported drier soil under spruce           |
|     | but these studies covered only the upper part of the soil profile (0-30 cm). Viewed over two decades and the entire soil                |
| 475 | profile, the contrasting soil moisture regimes of individual studies prove to be precipitation-driven while differences between         |
|     | the forest types are dominated by depths of 30 cm and more, where the greatest differences arise. The latter finding                    |
|     | highlights the need for soil moisture measurement at greater depths, which are too often neglected.                                     |
|     | We also found a surprising trend of intra-annual precipitation redistribution in the catchment since 1975. Our observations             |
|     | show significantly decreasing winter (slope -1.7 mm y -1 , p-value 0.061) and insignificantly increasing summer P (slope 1.7 |
| 480 | mm y -1 , p-value 0.24), which is entirely contrary to prevailing expectations based on climate model predictions (Kyselý et |
|     | al., 2011). Given that seasonal P sums interact with vegetation processes to affect the overall water balance, the actual               |
|     | direction of this trend will not only determine when water arrives in the system but also how it is partitioned.                        |
|     | A precipitation shift in either direction would reinforce the ecohydrological differences between the two forest types. With a          |
|     | shift to winter precipitation, differences in summer transpiration would be abated by lower growing-season input and                    |
| 485 | groundwater recharge would become increasingly reliant on deciduous forest due to its low winter interception. By contrast,             |
|     | a shift to summer precipitation would decrease ET from winter interception by evergreen forest and increase the importance              |
|     |                                                                                                                                         |

site. This can be explained by species-specific plant hydraulic traits. Beech has a wider and deeper rooting pattern and thus

20

of montane spruce forest to recharge. The rates of groundwater recharge under the two forest types will thus continue to

Further developments in the forest species composition of montane catchments is also likely to play a role. Given the present

dominance of spruce, the precipitation seasonality trend in our catchment is consistent with groundwater recharge shifting to

diverge under increased water limitation in either precipitation seasonality scenario.

490

4.3 Scope of the study By focusing on a pair of highly instrumented sites in a long-term experimental catchment, our study design allows the key processes to be examined in detail at the appropriate scale at the expense of broad landscape representativeness.

these interactions will become increasingly important to detailed projections of water flux partitioning.

495

the summer and offsetting overall drying somewhat. Increasing representation of beech would exacerbate higher atmospheric demand, given their ability to consume soil water even during drought periods. A combined trend of wetter winters and increasing representation of beech trees in Central Europe, would lead to even higher winter groundwater recharge and runoff. Overall, these various possible trajectories underscore the key role of climate-vegetation feedbacks in modulating

how hydroclimatic changes actually affect water balance. Given ongoing hydroclimatic shifts, process understanding of

500 Nevertheless, the landscape position of the study system gives it particular significance to projections of future ecological and hydrological dynamics across the region. Through both locally higher inputs and intra-annual storage, forested montane

2007, Immerzeel et al. 2020). The broader landscape's (i.e., downstream) water regimes will be particularly sensitive to their seasonal functioning under climate change. Furthermore, the observation of an annual-scale switch from energy- to a waterlimitation in a montane forest is strongly indicative for large parts of the generally warmer, drier Central European 505

headwater catchments play an outsize role in baseflow generation, supporting regional hydrological stability (Viviroli et al.,

landscape. On the other hand, the resulting process understanding is only transferable to an extent circumscribed by an adequate

consideration of the landscape position of the study system. For example, while summer season temperature did not greatly affect the water balance in our study catchment, this may in part be due to comparatively low a vapour pressure deficit

510 (VPD) at this elevation. As VPD is a strongly nonlinear function of air temperature (Groissord et al., 2021), lower elevation forests will face disproportionately higher summer VPD, potentially increasing its importance in their water balance. This factor may also increase in importance disproportionately across the landscape with further climate warming. Given our

findings, we would again expect any increased effects to be stronger in beech rather than spruce stands, due to their relatively anisohydric transpiration, and to shift the state of these systems further towards water limitation. It should urgently be evaluated how widespread the observed deviation from the predicted trend in the seasonal timing of

515 precipitation is. If it is merely a strong local anomaly, we would expect drier summers to exacerbate overall water-limitation and the importance of winter recharge from deciduous forest to increase over time. If the trend we found is real but limited to

catchments with the specific landscape position of ours (e.g., similar exposure and position within the Bohemian Forest),

then these catchments may play an offsetting role in the shifting regional water balance, smoothing out shifts in recharge. If, on the other hand, this deviation is due to a general (e.g., orographic) effect not accounted for in climate models, it may 520 generalise to the entire Bohemian Forest and reverse expectations about both the seasonality of future water availability and,

through interactions with vegetation, its annual sums in the region.

**4.4 Measurement limitations**

As the measuring limit of the tensiometers is -865 cm (-85 kPa), pressure heads below this limit could not be recorded.

Some information was therefore lost, especially at the beech site where periods with a constant limit value were clearly visible. However, for pressure heads lower than the measurement limit, the loss and gain of the volumetric water content corresponding to the unit change in the pressure head is very small (a 100 cm change in the pressure head accounts for less

corresponding to the unit change in the pressure head is very small (a 100 cm change in the pressure head accounts for less than 0.002 cm3 cm-3 of the change in the volumetric water content). The same rate was observed for a saturation to a pressure head of -100 cm, which is equal to 0.22 cm3 cm-3. Hence, the changes in pressure head concerning such low heads have a negligible effect on the volumetric soil water content.

To encompass the influence of soil moisture spatial variability, 2 to 5 tensiometers were used at each depth. As the standard

errors of precipitation measurements are 10% in summer and 40% in winter, it can be assumed that these measurements of precipitation can also be biased due to wind eddies around the rain gauge and deposited precipitation (Dingman, 2015). Even though the study sites are located close to the rain gauges (<500 m) and we also checked the open area rainfall data with the

though the study sites are located close to the rain gauges (<500 m) and we also checked the open area rainfall data with the raingauges located in the forest, there were occasional episodes in the data where the volumetric water content did not match to the volume measured rainfall, which resulted in a few errors in the soil moisture modelling, especially of the rises in the volumetric soil moisture content.

**4.5 Modelling limitations**

Observations from the above-mentioned periods when soil pressure heads were at or below the measuring range of the tensiometer were not used to constrain or evaluate the soil water balance model. The model was allowed to run freely below

this limit, and the error statistics from these periods were not considered. Eliminating this bias did not allow model fitting during dry periods. Another issue arose from the noted episodes of rainfall over-andunderestimation. As both issues affected periods with negligible water fluxes, neither was found to affect the long-term water balance. Finally, as shown in Cejpek et al. (2018) and Jačka et al. (2021), different vegetation species growing on the same soil type tend to change soil

properties, whether due to different root systems, soil biology or litter. Even though the soil parameters (Ks,  $\Theta_{e,r,s}$ ) that were entered into the balance model have measured equivalents at each site, their values in this study are the result of model calibration.

calibration.

As we used a simple temperature-based approach for the estimation of PET we compared these estimates with state-of-theart method of Penman-Monteith (Monteith, 1965) over the period of available data (from 2008). The influence of method

art method of Penman-Monteith (Monteith, 1965) over the period of available data (from 2008). The influence of method selection on the resulting water fluxes was negligible (<2% on seasonal and annual PET, AET, modelled soil water content). The sensitivity of PET to canopy-specific aerodynamic resistance parameterisation (beech vs spruce) in the Penman-

Montieth approach was in our case outweighed by the influence of soil water availability (reflected in stomatal resistance).

mostly covered by spruce forest, it is possible that these values may not correspond with the discharge that might occur from the beech site alone. This might affect confidence in the balance components (drainage and actual evapotranspiration) at the beech site as compared to the spruce site. However, the modelled high transpiration rates at the beech sites mostly follow

We limited our inferences to seasonal and annual comparisons, at which scales the differences between the PET estimation

Since the model validation was performed on the average annual discharge value measured for the entire watershed, which is

from fitting to the high-resolution time series of measured local soil moisture data, which show lower values during the summer season compared to spruce, and simultaneous observations of no change in groundwater levels. The higher modelled transpiration rates of beech during the summer season presented in this study could also be supported by the higher measured sap flow during the summer season in Switzerland (Brinkmann et al., 2016) or nearby Kranzberg forest in Bavaria (Gebhardt et al., 2023). Moreover, due to the absence of measured soil moisture data below the tensiometer measurement limit, it could be assumed that as soil moisture values could be even lower at beech sites, transpiration will be higher than estimated. To

avoid such uncertainties in future research, detailed sap flow measurements might serve for model calibration, which could

**then show the values of actual evapotranspiration and drainage more precisely. 570 **5 Conclusion**

585

methods are negligible.

**Ongoing climate change is forcing a transition from energy- to water-limitation and altering the species composition of European forests. We analysed a multi-decade record of soil water potential and climatic data to determine which variables have driven water limitation so far and which vegetation processes most exacerbate or dampen it. We found evidence of annual-scale water limitation, unprecedented in Central European montane forest. While increasing atmospheric demand**

annual-scale water limitation, unprecedented in Central European montane forest. While increasing atmospheric demand

drives progressive water limitation at the broader scale, seasonal water supply interacts with vegetation processes to

determine the actual soil water balance in the studied beech and spruce stands. Decreased summer precipitation drove

stronger drying in the beech stand compared to spruce. Our water-balance model suggests that beech did not reduce

transpiration rates in dry summers but continued to exploit deeper soil water reserves more extensively (by ~60 mm season-1 on average), resulting in decreased drainage. During wet summers and all winter seasons, the soil was drier in the spruce stand, due to higher winter interception by its evergreen canopy (by ~40 mm season-1 on average). Hence, in wet periods,

stand, due to higher winter interception by its evergreen canopy (by ~40 mm season-1 on average). Hence, in wet periods, drainage remained higher in the beech forest.
 The results suggest that with progressing water-limitation, soil water will increasingly be disproportionately depleted in by

forests composed of deeper-rooted, more anisohydric species. The combined effects of climate and forest composition change may thus increase the severity of summer soil drought and limit groundwater recharge. On the other hand, increasing

the proportion of deciduous species should result in increased winter recharge, due to decreased interception by leafless

canopies. As climate-vegetation interactions represent key sources of uncertainty in predicting shifts in ecosystem function and composition under climate change, we expect such advances in process understanding will contribute to the next

**Competing interests** The authors declare that they have no conflict of interest. Acknowledgement This research was supported by the Czech Science Foundation (GA CR 24-10375S), the institutional support of the Czech

**600**

shift.

**Author contribution**

manuscript.

590

595

Academy of Sciences, Czech Republic (RVO: 67985874, 67985939), and by the programme framework of the Strategy AV21. The authors would like to thank David Pesta for conducting regular field measurements.

References Arnold, J. G., Kiniry, J. R., Srinivasan, R., Williams, J. R., Haney, E. B., amd Neitsch, S. L.: Soil and Water Assessment Tool Input/Output Documentation: Version 2012. Texas Water Resources Institute, College Station, 2012.

generation of models and projections, facilitating both ecosystem and water management during the ongoing hydroclimatic

VS, MT, LV created the concept and set the methodology; NZ, VS, JT, JK carried the investigation; NZ implemented the model and created visualizations; NZ, VS and MB wrote the manuscript draft; all authors reviewed and edited the

605

Braden, H.: Ein Energiehaushalts-und Verdunstungsmodell for Wasser und Stoffhaushaltsuntersuchungen landwirtschaftlich genutzer Einzugsgebiete, Mittelungen Deutsche Bodenkundliche Geselschaft, 42, 294-299, 1985.

Brázdil, R., Dobrovolný, P., Mikšovský, J., Pišoft, P., Trnka, M., Možný, M., and Balek, J.: Documentary-based climate reconstructions in the Czech Lands 1501-2020CE and their European context, Clim. Past, 18, 935-959, doi: 10.5194/cp-18-935-2022, 2022.

610

Brázil, R., Zahradníček, P., Dobrovolný, P., Štěpánek, P., and Trnka, M.: Observed changes in precipitation during recent warming: The Czech Republic, 1961–2019, Int. J. Climatol., 41, 3881–3902, doi:10.1002/joc.7048, 2021. Brinkmann, N., Eugster, W., Zweifel, R., Buchmann, N., and Kahmen, A.: Temperate tree species show identical response in

[revised manuscript text omitted]

A. C., Emmer, A., Feng, M., Fernández, A., Haritashya, U., Kargel, J. S., Koppes, M., Kraaijenbrink, P. D. A.,

predict shifts in montane forest carbon–water relations, PNAS, 115(18), E4219–E4226, doi:10.1073/pnas.1718864115, 2018. Mianabadi, A., Davary, K., Pourreza-Bilondi, M., Coenders-Gerrits, A.M.J.: Budyko framework; towards non-steady state conditions, J. Hydrol., 588, 125089, doi: 10.1016/j.jhydrol.2020.125089, 2020. 705 Milly, P. C. D., Betancourt, J., Falkenmark, M., Hirsch, R. M., Kundzewicz, Z. W., Lettenmaier, D. P., Stouffer, R. J., Dettinger, M. D., and Krysanova, V.: On critiques of "Stationarity is dead: Whither water management?", Water

125390, doi:10.1016/j.jhydrol.2020.125390, 2020.

Resour. Res., 51, 7785–7789, doi:10.1002/2015WR017408, 2015.

doi:10.1016/j.jhydrol.2011.11.057, 2012.

332, doi:10.1007/s10342-023-01628-y, 2024.

675

680

685

690

695

700

710

715

720

2021.

2011.

Hydrol.,

420-421.

[revised manuscript text omitted]

package, Hydrol. Earth Syst. Sci., 16, 3315–3325, doi:10.5194/hess-16-3315-2012, 2012.

Tolasz, R., et al.: Climate Atlas of Czechia. Czech Hydrometeorological Institute, Prague, 2007.

725

730

735

740

745

750

765

770

2020.

Viviroli, D., Dürr, H. H., Messerli, B., Meybeck, M., Weingarten, R.: Mountains of the world, water towers for humanity: Typology, mapping, and global significance. Water Resour. Res., 43, 1–13, doi:10.1029/2006W R0056 53, 2007. von-Hoyningen-Huene, J.: Die interzeption des Niederschlages in landwirtschaftlichen Pflanzenbeständen, DVVVK, 57, 3–

the Czech Republic, Hydrol. Earth Syst. Sci., 21(2), 963–980, doi: 10.5194/hess-21-963-2017, 2017.

- 10.1080/17445647.2013.800827, 2013.
- Vondráková, A., Vávra, A., Voženílek, V.: Climatic regions of the Czech Republic, J. Maps, 9(3), 425-430, doi: 760 Wang, H., Tetzlaff, D., and Soulsby, C.: Modelling the effects of land cover and climate change on soil water partitioning in a boreal headwater catchment, J. Hydrol., 558, 520-531, doi:10.1016/j.jhydrol.2018.02.002, 2018.

Yue, S., Pilon, P., Phinney, B., Cavadias, G.: The influence of autocorrelation on the ability to detect trend in hydrological

Zahradníček, P., Brázdil, R., Štěpánek, P., and Trnka, M.: Reflections of global warming in trends of temperature characteristics in the Czech Republic, 1961–2019, Int. J. Climatol., 41, 1211–1229, doi: 10.1002/joc.6791, 2020. Zucco, G., Brocca, L., Moramarco, T., and Morbidelli, R.: Influence of land use on soil moisture spatial-temporal variability and monitoring, J. Hydrol., 516, 193–199, doi:10.1016/j.jhydrol.2014.01.043, 2014. Zweifel, R., Böhm, J. P., and Häsler, R.: Midday stomatal closure in Norway spruce - Reactions in the upper and lower

**Supplementary Material**

Figure S1: Scheme representing the workflow of the study

Table S2: Average soil hydraulic parameters of all soil layers derived from direct measurements. Each value is depicted by its mean  $\pm$  standard deviation.

| Measured SHP |          | $	heta_r$       | $\Theta_s$      | $\alpha$        | n               |
|--------------|----------|-----------------|-----------------|-----------------|-----------------|
|              | 10 cm    | $0.32 \pm 0.03$ | $0.70 \pm 0.04$ | $0.04 \pm 0.01$ | $2.10 \pm 0.53$ |
| Comica       | 35–45 cm | $0.18 \pm 0.04$ | $0.52 \pm 0.03$ | $0.04 \pm 0.01$ | $1.72 \pm 0.15$ |
| Spruce       | 50 cm    | $0.15 \pm 0.02$ | $0.48 \pm 0.02$ | $0.05 \pm 0.01$ | $1.58 \pm 0.19$ |
|              | 70–75 cm | $0.15 \pm 0.04$ | $0.50 \pm 0.04$ | $0.07 \pm 0.03$ | $1.45 \pm 0.13$ |
|              | 10 cm    | $0.17 \pm 0.02$ | $0.53 \pm 0.02$ | $0.05 \pm 0.01$ | $1.36 \pm 0.02$ |
| Beech        | 30 cm    | $0.18 \pm 0.01$ | $0.49 \pm 0.04$ | $0.05 \pm 0.01$ | $1.55 \pm 0.21$ |
| Beech        | 45 cm    | $0.17 \pm 0.01$ | $0.47 \pm 0.01$ | $0.05 \pm 0.01$ | $1.46 \pm 0.01$ |
|              | 60 cm    | $0.13 \pm 0.02$ | $0.42 \pm 0.01$ | $0.05 \pm 0.02$ | $1.48 \pm 0.13$ |

Figure S3: Model performance when calibrated in particular periods. Values from first columns represent calibration from 2000 to 2004, the second and following columns represent the following calibration periods (2005-2009, 2010-2014, 2015-2019, and the last column is an overall calibration)

Figure S4: Measured and modelled snow water equivalent (a) and soil water content (b)

---

## Author Response (AR2)

Divergent water balance trajectories under two dominant tree species in montane forest catchment shifting from energy- to water-limitation

Nikol Zelikova et al.

**Author's response to Reviewer#1**

**Comment #1**

There seems to be no consideration of the difference in the phenological phases between the beech stand and the spruce stand and their influence on the transpiration.

**Response #1**

Yes, you are right. The utilized modelling approach did not take into account phenological phases of particular vegetation. Hence, we have newly incorporated crop growth phases based on the seasonal variation of leaf area index in the beech forest. The approach originates from the FAO56 manual and adjusts the rate of potential evapotranspiration based on the crop coefficient ( $K_c$ ). In our case, it is expressed as a ration of LAIactual to LAImax, hence varying from 0 to 1. This enables the gradual increase in transpiration of beech with the onset of growing season as the PET will be multiplied by increasing value of  $K_c$ .

It inevitable led to the new calibration of the model at the beech site (new model parameters in Table 2), which resulted in slightly different values of water fluxes and therefore changes in Table 3 and Figure 6,7 and 8. The seasonal values remained nearly the same but there is more pronounced difference in summer/winter rates of transpiration and drainage between both sites.

**Comment #2**

Since data for the Penman-Monteith (P-M) method has only been available at this location since 2008, it might be reasonable to extend the time series with this simple approach. However a comparison with a more refined method should be shown in the manuscript or at least in the supplementary.

**Response #2**

We newly present the comparison with Penman-Monteith approach (that was a part of responses to reviewers) in the supplementary material (Fig S6).

**Comment #3**

The relative homogeneity of vegetation is not a strong indicator for the representativity of the soil moisture measurements. The convenient placement of the sensors is also only a necessary condition but not sufficient. Soil structure, particularly in soils with a high skeleton content, is much more important for water drainage, root penetration and finally plant-available water.

The additional measurements with the UMS T8 indicate a reasonable correlation with the Thies sensors. Please, put the graphics also in the supplement.

**Response #3**

Unfortunately, I am afraid that we cannot add the required plot into supplementary material as it was already published in another journal (eventhough also in the supplementary material). Hence, in this case, we would prefer to keep only the reference to our previous manuscript where the plot is easily accessible.

**Comment #4**

I still miss a graphical presentation of the soil moisture changes observed over time. Could you add the average values of soil moisture and runoff to Figure 2.

**Response #4**

We have newly modified Fig 2 so that it contains average seasonal soil water content and runoff.

**Comment #5**

Thank you for the exceedance probabilities of pressure head (Fig. 3). I am wondering why the probability for the entire period does not run in between the dry and the wet years. Why is the pressure head by 100 % probability of exceedance not the same for entire period and for wet years? Likewise, why is the pressure head by 0 % probability of exceedance not the same for entire period and for dry years? This should be the minimal and maximal value contained in both datasets respectively.

**Response #5**

Thank you for the thoughtful comment, we have constructed the average exceedance probabilities for the entire period in the wrong way. Originally, we made an average daily pressure head for each particular day (as it is usually done in streamflow analysis) instead of taking the entire dataset in consideration without averaging, which lead to the fact that the probabilities for the entire period did not run in between the dry and wet years. Now it is constructed correctly so that the exceedance probabilities for the entire period run in between wet and dry years and similarly pressure head of 0 %/100% probability of exceedance is the same for entire dataset as for dry/wet years.

**Comment #6**

Please, place a concise description of the categories directly after L293 "... seasonal development of their measured pressure heads.", i.e. move the part after L314 up to L293 and complete it with values (L314 is not a sum up, but a qualitative definition of the categories).

**Response #6**

The description was moved.

**Comment #7**

- The description of the model parameterisation and validation has improved with subsection 2.4, but it still needs some clarification. Please, make short sentences and use tables or lists for description of parameters. Clearly indicate which parameters are calibrated using which data and quality criteria.
- If you use the runoff for calibration you need to define the areal distribution of beech and spruce in the catchment.
- L230 You state "4 sub-periods for cross-validation". Cross-validation is a statistical technique used to evaluate how well the results of a model or analysis will generalize to an independent, unseen dataset. It involves partitioning the available data into subsets, training the model on some of these subsets, and testing it on the remaining ones. This process is repeated multiple times to ensure that the model's performance is robust and not just tailored to a specific portion of the data. ... It is not clear what is the unseen dataset which you use for the cross-validation.
- I would assume that there is a difference in the parameter set for summer and winter, at least at the spruce site.
- L240: "forward modelling"? Did you start with a known model and predict observations? It is more "inverse modelling", where you derive the model parameter, i.e. the model, from observations.
- L241 "Besides the model run in the period of available soil water potential measurements (2000–2021), the model was run also from 1975 to 1999 using the calibrated model parameters and available air temperature and precipitation sums to quantify annual AET for the entire observation period." This description is not really reproducible.

**Response #7**

- The first paragraph of the 2.4 section was rewritten and the list was model parameters was newly added to Table 1.
- The area distribution of beech and spruce is mentioned in the 2.1. Study site section (line 114)
- We have newly made clearer what were those unseen datasets in line 241-242
- Although, it is a very tempting approach, which we have already explored (Sipek and Tesar, 2017) we did not use different parameter sets for summer and winter in this study.
- It was a improper phrase, which was deleted.
- The section was rewritten (lines 236–239).

**Comment #8**

How is the influence of tree type regarded?

You respond "The influence of tree type is reflected through different parametrization of the effective wetness (theta\_E) restricting the rate of PET." However, the parameters theta\_S and theta\_R depend only on the soil type and soil structure, not on the tree type.

**Response #8**

You are absolutely right that the mentioned parameters depend on the soil type, but in reality, they also reflect the vegetation properties as the slope of the PET reduction in Feddes function (given by the difference in ThetaS and ThetaR) varies among both sites. Additionally, as we implemented the crop growth phases based on the seasonal variation of leaf area index in beech forest (which influence both transpiration and interception), the tree type influence is incorporated in the model.

**Comment #9**

These are indeed a complex relationships; one could also investigate the dependence on the soil moisture of the previous year or the runoff.

**Response #9**

Yes, we did that originally as well but the correlation was very low (correlation coefficient was -0.11) as for the snow cover and air temperature; hence we did not include the information.

**Comment #10**

Fig. 7: Budyko plots are interesting. Please, compare the slidgly changed form in Renner et al. (2014). in my opinion it is somewhat clearer.

It is also to consider that in this study PET is only a linear function of the air temperature. A trend in PET is therefore initially a trend in temperature. AET is then the relative filled soil water storage times "Temperature" plus interception.

For PET/P < 1 the system is just energy limited, crossing the line of PET/P = 1 the system is additionally water limited. Instead of Fig 5 a) with the 5 year sums I would prefer a time series of AET/PET it might be that the shift between the two spaces in time is better visible using this relation.

**Response #10**

Thank you for the reference to the article of Renner et al. (2014) which we did not know. We newly modified the Fig 7a in order to express AET in the relation with PET as it better represents the long-term shift from energy to water limited environment (than the 5y averages) by the decline of AET/PET ratio and occurring divergence between beech and spruce site.